# HWC-Loco: A Hierarchical Whole-Body Control Approach to Robust Humanoid Locomotion

**Sixu Lin[1], Guanren Qiao[1], Yunxin Tai[2], Ang Li[1], Kui Jia[1], Guiliang Liu[1,3,∗]**
[1]School of Data Science, The Chinese University of Hong Kong, Shenzhen,
[2]DexForce Technology, [3]Shenzhen Loop Area Institute.

## Abstract

Humanoid robots, capable of assuming human roles in various workplaces, have become essential to embodied intelligence. However, as robots with complex physical structures, learning a control model that can operate robustly across diverse environments remains inherently challenging, particularly under the discrepancies between training and deployment environments. In this study, we propose HWC-Loco, a robust whole-body control algorithm tailored for humanoid locomotion tasks. By reformulating policy learning as a robust optimization problem, HWC-Loco explicitly learns to recover from safety-critical scenarios. While prioritizing safety guarantees, overly conservative behavior can compromise the robot's ability to complete the given tasks. To tackle this challenge, HWC-Loco leverages a hierarchical policy for robust control. This policy can dynamically resolve the trade-off between goal-tracking and safety recovery, guided by human behavior norms and dynamic constraints. To evaluate the performance of HWC-Loco, we conduct extensive comparisons against state-of-the-art humanoid control models, demonstrating HWC-Loco's superior performance across diverse terrains, robot structures, and locomotion tasks under both simulated and real-world environments.

## 1 Introduction

Humanoid robots, with a physical structure resembling that of humans, can be seamlessly integrated into humans' workspace and take on roles in completing different tasks Saeedvand et al. (2019). This capability makes them reliable embodiments of artificial intelligence in various forms Paolo et al. (2024). In recent years, with advancements in both hardware capabilities and control algorithms, humanoid robots have become an increasingly significant type of robot with a growing impact on practical applications across various environments, such as factories,

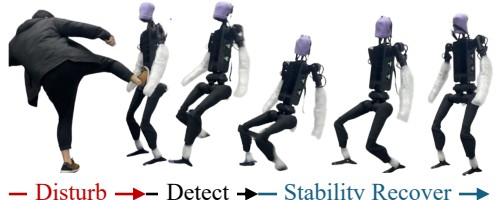

— Disturb ➤— Detect ➤— Stability Recover ➤

Figure 1: An example of recovering from a Hard Kick: The humanoid robot withstands external disturbance by automatically detecting hazardous states and adjusting its motion to regain stability.

homes, and offices (Nasiriany et al., 2024; Tao et al., 2024; Zhang et al., 2025b).

In the development of humanoid control, classic methods commonly rely on model-based optimization (Sakagami et al., 2002; Sentis & Khatib, 2006; Gouaillier et al., 2009; Radford et al., 2015; Chignoli et al., 2021). However, these methods require precise and comprehensive modeling of the robot's structure, kinematics, and dynamics across different environments. In practice, obtaining such data requires significant computational resources or manual effort. These limitations significantly influence the scalability of these approaches. To develop an end-to-end solution with promising generalizability, recent studies (Zhuang et al., 2024; Radosavovic et al., 2024; Long et al., 2024; He et al., 2024b; Gu et al., 2024c; Qiao et al., 2025) adopted Reinforcement Learning (RL) methods by training a neural model to control based on human demonstrations and interactions with the environment.

---

∗Corresponding author: Guiliang Liu, liuguiliang@cuhk.edu.cn.

While these learning-based approaches show potential for effective control policies across various tasks, they are typically trained in simulated environments Qiao et al. (2023), which often differ considerably from the deploying environment. To bridge the Simulation-to-Real (Sim2Real) gap, previous works often introduce additional regularization to constrain the robot's movements or employ domain randomization to account for variations in the robot's physical properties (Lai et al., 2023; Rudin et al., 2022; Nahrendra et al., 2023). However, excessive regularization can greatly affect the efficiency of control policy, and unstructured randomization often fails to capture safety-critical patterns in real-world applications.

To develop a reliable locomotion policy capable of generalizing from the training to the deployment environment, we propose formulating policy optimization as a robust optimization problem under misspecified environmental dynamics. However, traditional robust optimization primarily focuses on worst-case control, requiring the policy to ensure safety under the most adverse environmental dynamics within an uncertainty set. Under this context, the learned policy tends to be overly conservative, inducing sub-optimal control performance in terms of achieving the given task.

In this paper, to learn a control policy that can dynamically solve the trade-off between task performance and maintaining safety under different deployment environments, we present **H**ierarchical **W**hole-body **C**ontrol for Robust Humanoid **Loco**motion (HWC-Loco), where a high-level planning policy switches between a goal-tracking policy for task execution and a safety-recovery policy for stability under disturbances. The goal-tracking policy ensures task performance and human-like motion patterns, while the safety-recovery policy enforces ZMP-based constraints under extreme scenarios. A high-level planner coordinates these policies by dynamically selecting the appropriate one based on the deployment situation. Figure 1 illustrates an example of HWC-Loco, where a *high-level planning policy* detects hazardous states and seamlessly switches from the *goal-tracking policy* to a *safety-recovery policy*.

We conduct extensive experiments to evaluate the performance of HWC-Loco from four fundamental perspectives: 1) *effectiveness*, demonstrated across diverse terrains, 2) *robustness*, evaluated under varying scales of disturbances, 3) *naturalness*, exhibited in its ability to imitate human behavioral norms, 4) *scalability* evaluated in different robot embodiment and motion tracking tasks. These validations are performed in both simulated and real-world settings, demonstrating HWC-Loco as a foundational advancement for achieving reliable humanoid locomotion in safety-critical scenarios.

## 2 RELATED WORK

**Learning-based Legged Robot Locomotion.** Various learning-based methods have been proposed for legged locomotion, such as quadrupedal locomotion (Rudin et al., 2022; Shi et al., 2022; Chen et al., 2023; Nahrendra et al., 2023; Margolis & Agrawal, 2023; Zhuang et al., 2023; He et al., 2024d) and bipedal locomotion (Li et al., 2021; Kumar et al., 2022; Li et al., 2023; Duan et al., 2024). Despite their success, these approaches cannot be directly applied to humanoid robots because of the complex physical structure and higher degrees of freedom (DOF). Recent advances in hardware and learning methods have enabled humanoid robots to move through diverse environments. Some humanoid control policy navigates across diverse terrains using only proprioceptive information (Radosavovic et al., 2024; Gu et al., 2024b;c). Others enhance performance by incorporating additional sensors, such as vision or LiDAR, to collect detailed environmental data, enabling robots to perform complex tasks like stair climbing and parkour jumps (Zhuang et al., 2024; Long et al., 2024; Sun et al., 2025). While these approaches demonstrate promising locomotion performance, they lack mechanisms to handle safety-critical scenarios humanoid robots may encounter during real-world deployment.

**Whole-body Control for Humanoid Robots.** Learning effective whole-body control remains a central challenge for humanoid robotics. Prior works (Luo et al., 2023a; 2024a;b) have demonstrated impressive results in simulated avatars, showcasing the feasibility of learning-based control strategies. More recently, studies such as (Cheng et al., 2024; He et al., 2024b;a; Fu et al., 2024; He et al., 2024c; 2025; Ze et al., 2025; Li et al., 2025; Liao et al., 2025; Huang et al., 2025; Zhao et al., 2025) have enabled realistic robots to imitate complex human-like motions, such as boxing and dancing by leveraging motion priors from human demonstrations. Additionally, previous methods employed multi-stage reward designs to accomplish complex tasks (Kim et al., 2024; Zhang et al., 2024a). However, the success of these methods often relies on intricate reward shaping and struggles to generalize across different humanoid robots, particularly when motion priors are unavailable.

## 3 PROBLEM FORMULATION

**Learning Environment.** We formulate the environment of the humanoid locomotion task with a Partially Observable Markov Decision Process (POMDP) $M = (\mathcal{S}, \mathcal{A}, \mathcal{O}, P_\mathcal{T}, r, \mu_0, \gamma)$, where: 1) Within the state space $\mathcal{S}$, a state $s \in \mathcal{S}$ records the environmental information and the robot's internal states. 2) $\mathcal{A}$ denotes the action space, and action $a \in \mathcal{A}$ denotes the target joint angles that a PD controller uses to actuate the DOFs. 3) $o \in \mathcal{O}$ denotes an observation, which provides partial information about the state of the agent due to sensory limitations and environmental uncertainty. At a time step $t$, $\boldsymbol{o}_t$ includes velocity command $\boldsymbol{v}_t^{cmd} \in \mathbb{R}^3$ and proprioception $\boldsymbol{o}_t^p$ that records a humanoid's internal state. Appendix A.2 introduces the details. By incorporating the temporal observations $\boldsymbol{o}_t^H = [\boldsymbol{o}_t, \boldsymbol{o}_{t-1}, \cdots, \boldsymbol{o}_{t-H}]$, the control model can summarize a state as $s_t = [\boldsymbol{o}_t^H, \mathbb{P}_t]$, where $\mathbb{P}_t$ is the privileged information which the robot can't access in realistic deployment, including base velocity $v_t$, terrain height, external disturbance and ZMP features (see Section 4.3). 4) $P_\mathcal{T} \in \Delta_{\mathcal{S} \times \mathcal{A}}^{\mathcal{S}}$ denotes the transition function as a mapping from state-action pairs to a distribution of future states. 5) $r$ denotes the reward functions, which typically consist of penalty, regularization, and task rewards. These reward signals significantly influence the optimality of the control policy, for which we provide a detailed introduction in the following. 6) $\mu_0 \in \Delta^{\mathcal{S}}$ denotes the initial state distribution. 7) $\gamma \in (0, 1]$ denotes the discounting factor. The robot's policy stops at a terminating state $\tilde{s}$, and the corresponding terminating time is denoted as $T \in (0, \infty)$.

Under this POMDP, our goal is to learn a policy $\pi \in \Delta_\mathcal{S}^\mathcal{A}$ according to the objective as follows:

$$\max_\pi \mathcal{J}(\pi, M) = \max_\pi \mathbb{E}_{\mu_0, p_\mathcal{T}, \pi} \left[ \sum_{t=0}^\infty \gamma^t \left[ \beta_\mathfrak{T} r_\mathfrak{T}(s_t, a_t) + \beta_\mathfrak{P} r_\mathrm{P}(s_t, a_t) + \beta_\mathfrak{R} r_\mathfrak{R}(s_t, a_t) \right] \right] \quad (1)$$

In humanoid locomotion, the total reward is typically expressed as a weighted sum of task rewards $r_\mathfrak{T}$, penalty rewards $r_\mathrm{P}$, and regularization rewards $r_\mathfrak{R}$ (He et al., 2024b;a; Zhuang et al., 2024), each serving distinct purposes:

- *Task rewards $r_\mathfrak{T}$* evaluate goal achievement, e.g., velocity tracking Zhuang et al. (2024), contact management Zhang et al. (2024a), or expressive motions Cheng et al. (2024). They directly reflect policy optimality and thus serve as the main *objective to maximize*.

- *Penalty rewards $r_\mathrm{P}$* discourage undesirable outcomes such as falls or violations of torque and joint limits. Recent work (He et al., 2024b;a) assigns large weights $\beta_\mathfrak{P}$, effectively acting as Lagrange multipliers Gu et al. (2024a) but fixed rather than optimized. A more principled formulation is therefore a *constrained RL problem*.

- *Regularization rewards $r_\mathfrak{R}$* encourage motions aligned with human preferences for style or safety. Since such specifications are task-dependent and expert-driven Chen et al. (2024), we instead *align robot motion with human datasets* (e.g., AMASS Mahmood et al. (2019)), which embed rich behavioral priors.

**Constrained RL for Whole-Body Control.** Inspired by the aforementioned analysis, we formulate the whole-body control locomotion objective for humanoid robots as follows:

$$\max_\pi \mathbb{E}_{\mu_0, p_\mathcal{T}, \pi} \left[ \sum_{t=0}^\infty \gamma^t r_\mathfrak{T}(s_t, a_t) \right] \ s.t. \ \mathcal{D}_f(\rho_M^{\pi^E} \| \rho_M^\pi) \le \epsilon_f \ \text{and} \ \mathbb{E}_{\tau \sim (\mu_0, p_\mathcal{T}, \pi)} [\phi(\tau)] \le \epsilon_\phi \quad (2)$$

where 1) we replace the hard regularization in $r_\mathfrak{R}$ with a mimic learning objective $\mathcal{D}_f(\rho_M^{\pi^E} \| \rho_M^\pi)$ that constrains the distributional divergence between the learned policy's occupancy measure $\rho_M^\pi$ and the expert policy's occupancy measure $\rho_M^{\pi^E}$. In this study we implement $\mathcal{D}_f$ by Wasserstein distance (Section 4.1). 2) Instead of relying on the penalty reward $r_\mathrm{P}$, we use $\phi$ to capture the feasibility of the current trajectory generated by the policy $\pi$ under the environment $M$. Within the objective, only the task rewards $r_\mathfrak{T}$ are subject to maximization; other desired characteristics of humanoid robots are captured by constraints learned from human motion datasets. More importantly, this approach effectively eliminates the need for regularization on the humanoid robot's upper or lower pose, significantly enhancing the learning of whole-body locomotion policies.

**Robust Locomotion in Humanoid Robot.** A common approach to learning humanoid locomotion policies involves training in a simulated environment before deploying the policies in a real-world setting. Due to the complexity of real-world environments, there is often a significant discrepancy between the training and deployment environments. In this study, we characterize this discrepancy by the mismatched POMDPs defined as follows:

**Definition 3.1.** (Mismatched POMDPs) We define a POMDP $M := (\mathcal{S}, \mathcal{A}, P_\mathcal{T}, \mathcal{O}, r, \mu_0, \gamma)$, to be mismatched with another POMDP $M' := (\mathcal{S}, \mathcal{A}, P'_\mathcal{T}, \mathcal{O}, r, \mu_0, \gamma)$ if they differ only in transition functions (i.e., $P_\mathcal{T} \neq P'_\mathcal{T}$). We define a set of mismatched POMDPs as $\boldsymbol{\mathcal{M}}_m = \{M^{P_\mathcal{T}}, M^{P'_\mathcal{T}}, \dots\}$, where their transition functions differ from each other.

Note that we follow previous studies Moos et al. (2022) and primarily focus on the mismatch in transition dynamics during learning and deployment. The uncertainty set of transition function Viano et al. (2022) is represented as:

$$\mathfrak{P}^L_\alpha = \{\alpha P^L_\mathcal{T} + (1-\alpha)\bar{P}_\mathcal{T}, \ \forall \bar{P}_\mathcal{T} \in \mathfrak{P}\} \tag{3}$$

where $\alpha$ specifies the scale of mismatch, $P^L_\mathcal{T}$ denotes the transition function in the learning environment and $\mathfrak{P} \subseteq \Delta^{\mathcal{S}}_{\mathcal{S} \times \mathcal{A}}$ represents the set of all candidate transition functions. While other factors, such as divergence in state-action spaces and initial state distribution $\mu_0$, can influence the performance during deployment, we find that the majority of factors affecting Sim2Real performance in humanoid locomotion tasks can be characterized by the mismatch in transition dynamics. For example, if $\bar{P}_\mathcal{T}$ denotes a Gaussian function, the underlying sensor noise can be captured by transitions in $\mathfrak{P}^L_\alpha$. Additionally, if $\bar{P}_\mathcal{T}$ denotes a projection from one spatial state to another, the variations of terrains in the deploying environment can be modeled by transitions in $\mathfrak{P}^L_\alpha$ Long et al. (2024).

To solve a robust RL problem, previous studies often consider a max-min objective $\max_\pi \min_{M \in \boldsymbol{\mathcal{M}}_m} \mathcal{J}(\pi, M)$ where $\mathcal{J}(\pi, M)$ denotes a standard RL objective (1). With this objective, while the agent can learn a conservative policy to ensure worst-case control performance, this policy often compromises the control performance, making the humanoid less effective at tracking the given commands. To address this issue, we focus on ensuring a worst-case feasibility constraint, thereby reducing the influence of mismatched POMDP on maximizing task rewards. Accordingly, we update the CRL objective (2) to the robust humanoid locomotion objective represented as:

$$\max_\pi \min_{\widehat{P}_\mathcal{T} \in \mathfrak{P}^L_\alpha} \mathbb{E}_{\mu_0, P^L_\mathcal{T}, \pi} \left[\sum_{t=0}^{\infty} \gamma^t r_{\mathfrak{T}}(s_t, a_t)\right] \ s.t. \ \mathcal{D}_f(\rho^{\pi^E}_{M^{P^L_\mathcal{T}}} \| \rho^{\pi}_{M^{P^L_\mathcal{T}}}) \leq \epsilon_f \text{ and } \mathbb{E}_{\tau \sim (\mu_0, \widehat{P}_\mathcal{T}, \pi)}[\phi(\tau)] \leq \epsilon_\phi \tag{4}$$

where 1) $\mathfrak{P}^L_\alpha$ represents the set of mismatched dynamics from the learning environment dynamics $P^L_\mathcal{T}$ (defined in 3) and 2) $\rho^\pi_M(s, a) = (1-\gamma)\pi(a|s)\sum_{t=0}^{\infty} \gamma^t p(s_t = s|\pi, M)$ defines the normalized occupancy measure of policy $\pi$ under the environment $M$. Intuitively, we aim to learn a robust control policy that ensures feasibility across all mismatched transition functions $\widehat{P}_\mathcal{T} \in \mathfrak{P}^L_\alpha$, thereby guaranteeing the safety of policy $\pi$. However, for the reward-maximizing and mimic learning objectives, we optimize the policy based on the transition dynamics in the learning environment $P^L_\mathcal{T}$, rather than concentrating solely on worst-case guarantees.

## 4 HIERARCHICAL WHOLE-BODY CONTROL FOR HUMANOID LOCOMOTION

In practice, developing an end-to-end solver for the robust locomotion problem (4) is challenging, as it involves multiple deployment environments and constraints that must be adhered to. To tackle this issue, we propose dividing the objective into two stages: goal-tracking and safety recovery. The goal-tracking policy maximizes rewards while mimicking human behaviors (see Section 4.1). When mismatched environmental dynamics pose a risk to the safety of the control policy, HWC-Loco switches to the safety recovery policy, which manages safety-critical events to ensure the feasibility of humanoid control (see Section 4.2). To dynamically identify the "sweet spot" for policy switching or activation, we introduce a high-level policy that coordinates the low-level policies based on historical observations and the robot's current status (see Section 4.3). Figure 2 illustrates the training pipeline of our HWC-Loco.

### 4.1 LEARNING GOAL-TRACKING POLICY FOR EFFICIENT HUMANOID LOCOMOTION

The goal-tracking policy primarily focuses on tracking the provided command within the learning environment efficiently and naturally. This is achieved by optimizing the following objective function:

$$\max_{\pi_1} \mathbb{E}_{\rho^{\pi_1}_{M^{P^L_\mathcal{T}}}} \left[\sum_{t=0}^{\infty} \gamma^t r_{\mathfrak{T}}(s_t, a_t)\right] \ s.t. \ \mathcal{D}_f(\rho^{\pi^E}_{M^{P^L_\mathcal{T}}} \| \rho^{\pi_1}_{M^{P^L_\mathcal{T}}}) \leq \epsilon_f \tag{5}$$

Note that compared to the complete objective (4), this goal-tracking objective focuses only on maximizing performance in the training environments' dynamics $P^L_\mathcal{T}$. Within this objective, the scale

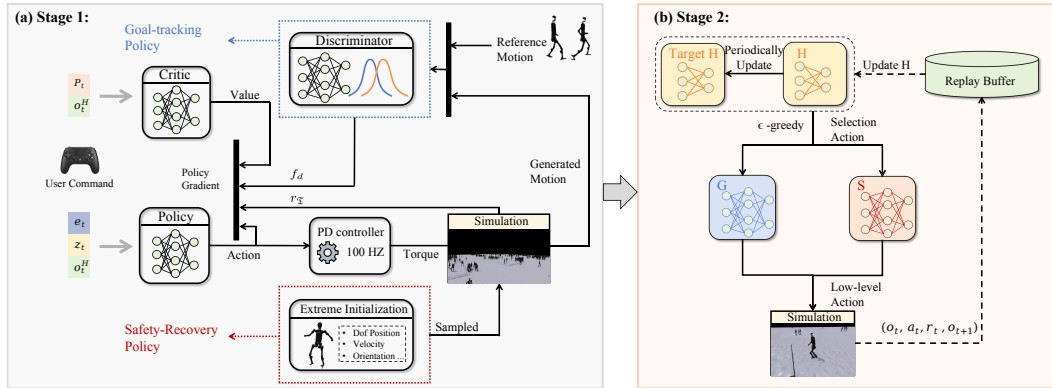

Figure 2: Overview of HWC-Loco: The framework consists of two stages: (a) Training *goal-tracking policy* to effectively enable human-like locomotion across diverse terrains (Section 4.1) and *safety recovery policy* to recover from safety-critical states (i.e., extreme-case) (Section 4.2). (b) Training the *high-level planning policy* to select between the two pre-trained low-level policies (Section 4.3), thereby ensuring stable and consistent locomotion.

of the cumulative task rewards, $\sum_{t=0}^{\infty} \gamma^t r_{\mathfrak{T}}$ reflects how effectively the policy $\pi$ tracks locomotion-related commands. Appendix A.2 specifies the configuration of our task reward.

In addition to task rewards, a crucial aspect of the goal-tracking objective (5) is evaluating the distance between the occupancy measures of the trained and expert policies. In the following, striving for concise representation, we represent $\rho^{\pi^E}_{M^P \mathcal{T}^L}$ and $\rho^{\pi}_{M^P \mathcal{T}^L}$ by $\rho^{\pi^E}$ and $\rho^{\pi}$. Driven by information theory, previous works Hussein et al. (2017) primarily implement $\mathcal{D}_f$ using metrics such as Kullback-Leibler (KL) divergence or Jensen-Shannon (JS) divergence. However, these metrics fail when the support of $\rho^{\pi^E}$ and $\rho^{\pi}$ has limited or zero overlap. In this study, we consider implement $\mathcal{D}_f$ with Wasserstein-1 distance under the Kantorovich-Rubinstein duality Villani et al. (2009) such that:

$$\mathcal{D}_f(\rho^{\pi^E} \| \rho^{\pi_1}) = \sup_{\|f_d\|_L \leq 1} \mathbb{E}_{x \sim \rho^{\pi^E}}[f_d(x)] - \mathbb{E}_{x \sim \rho^{\pi_1}}[f_d(x)] \quad (6)$$

where the bounded Lipschitz-norm $\|f_d\|_L \leq 1$ ensure the smoothness of functions $f_d$. Inspired by adversarial imitation learning (Ho & Ermon, 2016; Zhang et al., 2020), an effective and intuitive implementation of $f_d$ is a discriminator for measuring whether $x$ is generated by human experts, such that $f_d : \mathcal{X} \to \mathbb{R}$. It can be learned by maximizing the following objective:

$$\max_{f_d} \mathbb{E}_{x \sim \rho^{\pi^E}}[f_d(x)] - \mathbb{E}_{x \sim \rho^{\pi_1}}[f_d(x)] + \beta \cdot \mathbb{E}_{\hat{x} \sim [\rho^{\pi^E}, \rho^{\pi_1}]}\left[(\|\nabla_{\hat{x}} f_d(\hat{x})\|_2 - 1)^2\right] \quad (7)$$

where weighted by $\beta$, the gradient penalty term enforces the Lipschitz continuity required by the Wasserstein distance (i.e., $\|f_d\|_L \leq 1$). During training, $\rho^{\pi^E}$ represents the density of expert demonstration dataset $\mathcal{D}^E$, which is built from the CMU MoCap dataset cmu. Additionally, we include data about humans standing, walking, and running to learn various human behaviors. We then retarget the motion data into a humanoid robot-compatible format, which we use as the training dataset. By substituting $\mathcal{D}_f$ with equation (6), the goal-tracking objective (5) can be simplified as:

$$\max_{\pi_1} \mathbb{E}_{\rho^{\pi_1}}\left[\sum_{t=0}^{\infty} \gamma^t r_{\mathfrak{T}}(s_t, a_t)\right] \quad \text{s.t.} \quad \mathbb{E}_{\rho^{\pi^E}}[f_d(s^d)] - \mathbb{E}_{\rho^{\pi_1}}[f_d(s^d)] \leq \epsilon_f \quad (8)$$

where $s^d$ records the necessary state information for the discriminator. (See details in Appendix A.5). Given that this formulation of constrained RL problem has a zero duality gap Paternain et al. (2019), we can transform it into an unconstrained objective by analyzing the Lagrange dual form of the original constrained RL problem:

$$\max_{\pi_1} \mathbb{E}_{\rho^{\pi_1}}\left[\sum_{t=0}^{\infty} \gamma^t \left[r_{\mathfrak{T}}(s_t, a_t) - \lambda f_d(s^d)\right]\right] \quad (9)$$

where $\lambda$ denotes the optimal Lagrange multiplier. In practical implementation, the policy $\pi$ and discriminator $f_d$ can intuitively represent the generator and the discriminator under the adversarial

learning framework. In implementation, we utilize the proximal policy optimization algorithm Schulman et al. (2017) to update the goal-tracking policy. By alternatively updating (7) and (9), we can develop a goal-tracking policy that effectively maximizes task rewards while closely mimicking the expert's behavior.

## 4.2 LEARNING SAFETY RECOVERY POLICY FOR HANDLING SAFE-CRITICAL EVENTS

Our safety recovery policy primarily focuses on handling emergency events, thereby recovering the robot from safety-critical situations and preventing failures of locomotion tasks, such as loss of balance or control policy malfunctions. To learn the safety-recover policy, we mainly study the complete objective (4). Similarly, as it is introduced in Section 4.1, we implement the divergence metric $\mathcal{D}_f$ with Wasserstein-1 distance, deriving the following objective:

$$\max_{\pi_2} \min_{\widehat{P_\mathcal{T}} \in \mathfrak{P}_\alpha^L} \mathbb{E}_{\mu_0, \widehat{P_\mathcal{T}}, \pi_2} \left[ \sum_{t=0}^{\infty} \gamma^t r_{\mathfrak{T}}(s_t, a_t) - \lambda f_d(s_t, a_t) \right] \quad s.t. \quad \mathbb{E}_{\tau \sim (\mu_0, \widehat{P_\mathcal{T}}, \pi_2)} \left[ \phi(\tau) \right] \leq \epsilon_\phi \quad (10)$$

where $\epsilon_\phi$ is a hyperparameter. By comparing this objective with the goal-tracking objective (9), while both consider task rewards and human mimicking performance, their key differences lie in the incorporation of uncertainty in environmental dynamics and feasibility constraints in policy updates. These additions necessitate the construction of an extreme-case uncertainty set and the modeling of safety constraints by learning the feasibility function $\phi(\cdot)$.

**Extreme-case Uncertainty Set.** Real-world deployment has complex terrains, external disturbances, hardware malfunctions, and sensor noise. Those factors may lead to extreme cases like falling or unexpected behaviors. To model them in the simulation (i.e., modelling $\widehat{P_\mathcal{T}}$s), We: 1) apply multi-scale external forces and torques to the humanoid body, 2) introduce random, high-intensity noise to the proprioceptive information and PD gains, 3) resample velocity commands to simulate malicious velocity inputs Shi et al. (2024) within the goal-tracking policy, and 4) apply domain randomization to the training environment. Together, these dynamic variations construct the uncertainty set $\mathfrak{P}_\alpha^L$, under which we learn the safety-recover policy. Appendix A.6 shows our implementation details.

**Zero-Moment Point (ZMP) Constraint.** A bipedal robot can be generally modeled by a linear inverted pendulum (Morisawa et al., 2006; Harada et al., 2006), where ZMP refers to the point where the ground reaction force has no horizontal moment. If the ZMP exits the support polygon (typically the foot's contact area), the robot will quickly lose balance (Wieber, 2006; Feng et al., 2016; Scianca et al., 2020; Xie et al., 2025). Stability is governed by gravity and the Center of Mass (CoM) accelerations, which also account for disturbances like obstacles and slippery surfaces. Under this assumption, we implement the feasibility indicator in the safety recovery objective (10) as follows:

$$\phi(s, a) = \|\mathbf{p}_{\text{ZMP}}(s, a) - \mathbf{p}_{\text{ac}}\|_2 \quad \text{where} \quad \mathbf{p}_{\text{ZMP}}(s, a) = \mathbf{p}_{\text{CoM}}(s, a) - \frac{z_{\text{CoM}}(s, a)}{g} \cdot \ddot{\mathbf{p}}_{\text{CoM}}(s, a) \quad (11)$$

where 1) the vector $\mathbf{p}_{\text{CoM}}$ denotes the $x$ and $y$-coordinates of CoM position. 2) $z_{\text{CoM}}$ is the height of the CoM. 3) $g$ represents the acceleration due to gravity. 4) $\ddot{\mathbf{p}}_{\text{CoM}}$ indicates the accelerations of the CoM in the $x$ and $y$-directions. 5) $\mathbf{p}_{\text{ac}}$ represents the center of the support polygon (specific to each robot and known in advance). Intuitively, $\phi$ provides real-time insights into the robot's stability and varies in its representation depending on the current support phase. Therefore, the humanoid robot can assess its stability in real time and use whole-body coordination to satisfy the ZMP constraint.

## 4.3 HIGH-LEVEL PLANNING FOR POLICY TRANSITION

While the low-level policies can achieve varying levels of optimality, a fundamental challenge is coordinating these policies to ensure that a humanoid robot can efficiently complete tasks while adhering to safety and robustness requirements. We introduce a high-level planning policy that dynamically determines which policy to activate based on the specific state of humanoid robots. The task of policy selection has a discrete action space, and we learn this high-level policy $\pi_0$ based on the following objective:

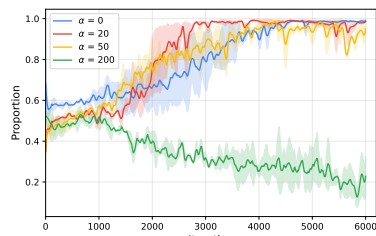

Figure 3: Average activation ratio of goal-tracking policy.

$$\max_{\pi_0} \mathbb{E} \left[ \sum_{t=0}^{\infty} \gamma^t \left[ r_{\mathfrak{T}}(s_t, \bar{a}_t) - \mathbb{1}(\bar{a}_{t-1} \neq \bar{a}_t) - \alpha \mathbb{1}(s_t) \right] \right] \quad (12)$$

where 1) the policy selection action $\bar{a}_t \in \{0,1\}^2$ denotes a one-hot vector featuring whether the goal-tracking policy $\pi_1$ and safety-recovery policy $\pi_2$, 2) $\mathbb{1}(\bar{a}_{t-1} \neq \bar{a}_t)$ denotes the continuality identifier for discouraging the over-frequent switch of policies, and 3) $\mathbb{1}(s_t)$ denotes the termination identifier for preventing the failing of locomotion tasks. (2) and (3) are both $r_P$ defined in Section 3. More importantly, within this objective, by adjusting $\alpha$, practitioners can adjust the trade-off safety guarantee and task completion. Training and algorithmic details can be found in Appendix A.2. We report the average activation ratio of $\pi_1$ during training in Figure 3. For $\alpha \in \{0, 20, 50\}$, $\pi_1$ dominates and $\pi_2$ is rarely invoked. In contrast, $\alpha = 200$ destabilizes training and induces overly conservative behavior due to sparse termination rewards.

**Practical Deployment.** The training of humanoid control policy commonly relies on the privileged information $\mathbb{P}_t$, such as external disturbances, terrain dynamics, etc. We follow Nahrendra et al. (2023) and infer $\mathbb{P}_t$ by training a VAE-based estimator $P(e_t, z_t | o_t^H)$ where $z_t$ captures the contextual embedding of the $\mathbb{P}_t$, and $e_t$ consists of body velocity and ZMP features. To enhance the representation of $\phi(\cdot)$, we introduce frequency encoding, similar to the approach in Mildenhall et al. (2021), to capture the subtle variations in $\phi(\cdot)$ for the policy. Additionally, instead of the zero-mean Gaussian prior, we introduce a learnable prior similar to Luo et al. (2023b). Moreover, we include a certain degree of domain randomization in the learning of HWC-Loco policies. The details of domain randomization are in Appendix A.6.

## 5 EXPERIMENT

**Experiment Settings.** To conduct a comprehensive evaluation, we quantify the control performance of HWC-Loco in simulated (Isaac Gym Makoviychuk et al. (2021)) and realistic environments from the following perspectives: 1) *Effectiveness*: How effectively does HWC-Loco navigate across diverse terrains? 2) *Robustness*: How well can HWC-Loco stabilize the humanoid robot under varying levels of disturbance? 3) *Naturalness*: How well does HWC-Loco imitate human-like movement norms? 4) *Scalability*: How effectively does HWC-Loco generalize to diverse embodiments and tasks?

Our evaluation metric includes a) *Success Rate*: The proportion of successful traverse across different scenarios. Cui et al. (2024) b) *Goal Tracking* performance: The ability to accurately follow velocity commands by maximizing task rewards $r_{\mathfrak{T}}$ detailed in Appendix A.2 Chen et al. (2024). c) *Human-Like* behavior: Measured as the Wasserstein-1 distance between the robot's and human motions. d) *ZMP Deviation*: The proportion of time the ZMP lies outside the support polygon.

To evaluate the effectiveness of different components in HWC-Loco, we compare against *domain-randomized policy* baselines to isolate the benefits of our hierarchical design: 1) *HWC-Loco-l* sets $\alpha$ to a lower value, thereby reducing the sensitivity of the high-level policy to failure events. 2) *Goal-Tracking Policy* removes the safety-recovery mechanism and relies solely on the goal-tracking policy $\pi_1$ (Section 4.1), which is trained under the same domain randomization as HWC-Loco. 3) *DreamWaQ-Humanoid* removes the human imitation objective $\mathcal{D}_f$ from objective (5), yielding a domain-randomized adaptation of DreamWaQ Nahrendra et al. (2023) for humanoid control. 4) To further highlight the advantages of our hierarchical policy, we compare against a recent *history-aware* extension of DreamWaQ, called *AHL* Cui et al. (2024), which employs two-phase training. Implementation and tuning details for all baselines are provided in Appendix A.7.

### 5.1 EFFECTIVENESS: LOCOMOTION ACROSS DIVERSE TERRAINS

**Effectiveness in Simulated Environments.** We evaluate the policies on challenging terrains, specifically slopes and stairs. To enable a comprehensive comparison, evaluations are performed under both low-speed and high-speed command conditions. Low-speed commands are sampled as $v_x^c \sim [0.0, 1.0]$, $v_y^c \sim [-0.3, 0.3]$, and $w_z^c \sim [-0.5, 0.5]$, while high-speed commands follow $v_x^c \sim [1.0, 2.0]$, $v_y^c \sim [-0.6, 0.6]$, and $w_z^c \sim [-1.0, 1.0]$. Here, $v_x^c$ and $v_y^c$ denote the commanded linear velocities along the x- and y-axes, respectively, and $w_z^c$ denotes the commanded angular velocity around the z-axis. Table 1 shows that our HWC-Loco achieves the highest success rate across all evaluated scenarios, while maintaining promising goal tracking and human mimic performance. Notably, in the high-speed stair terrain setting, HWC-Loco has a significantly higher success rate than all other policies, although its goal-tracking ability slightly decreases. This indicates that the policy prioritizes stability over aggressive goal pursuit. For example, when given a high-speed command, HWC-Loco enables the policy to adapt dynamically in safety-critical situations (e.g., navigating stairs mid-air) rather than rigidly maintaining high velocity. Comparably, when downplaying the sensitivity

to safety-critical events and removing the safety-recovery policy, the success data drops significantly from nearly **85%** to around **60%** in the testing environment with stairs and high-speed commands.

**Effectiveness in Realistic Environments.** Figure 10 (see Appendix B.6) demonstrates the robot's ability to climb 15 cm stairs and 20° slopes. When faced with challenging terrain, the robot prioritizes safety by activating the recovery policy. After regaining stability, it smoothly transitions back to goal tracking, showcasing its adaptive control in dynamic and uncertain environments. Figure 11 (see Appendix B.6) further illustrates the robot's effectiveness in real-world outdoor settings, where it successfully navigates across diverse terrains, including flat ground, grassy surfaces, and slopes.

Table 1: Locomotion performance in simulated environments. Each evaluation runs for 1200 steps, which is equivalent to 12 seconds of real clock time. $\pm$ corresponds to the standard deviation of the performance on 3 random seeds. The best result of each setting is marked as **bold**.

| Method | Slopes | | | Stairs | | |
|---|---|---|---|---|---|---|
| | Success Rate ↑ | Goal Tracking ↑ | Human - like ↓ | Success Rate ↑ | Goal Tracking ↑ | Human - like ↓ |
| **(a) Low Speed** | | | | | | |
| DreamWaQ | $92.31 \pm 0.40$ | $1.19 \pm 0.02$ | $3.52 \pm 0.05$ | $74.32 \pm 1.30$ | $1.19 \pm 0.03$ | $3.53 \pm 0.02$ |
| AHL | $98.83 \pm 0.15$ | $1.21 \pm 0.01$ | $3.41 \pm 0.04$ | $93.73 \pm 0.70$ | $1.21 \pm 0.04$ | $3.44 \pm 0.01$ |
| Goal-tracking | $99.90 \pm 0.04$ | $\mathbf{1.31 \pm 0.00}$ | $\mathbf{3.30 \pm 0.05}$ | $96.60 \pm 0.14$ | $\mathbf{1.31 \pm 0.00}$ | $3.31 \pm 0.01$ |
| HWC-Loco-l | $\mathbf{100.00 \pm 0.00}$ | $1.23 \pm 0.02$ | $3.32 \pm 0.03$ | $99.80 \pm 0.08$ | $1.21 \pm 0.02$ | $3.31 \pm 0.06$ |
| HWC-Loco | $\mathbf{100.00 \pm 0.00}$ | $1.22 \pm 0.02$ | $3.32 \pm 0.05$ | $\mathbf{99.98 \pm 0.01}$ | $1.19 \pm 0.03$ | $3.34 \pm 0.03$ |
| **(b) High Speed** | | | | | | |
| DreamWaQ | $90.46 \pm 0.43$ | $1.05 \pm 0.01$ | $3.63 \pm 0.04$ | $60.58 \pm 0.64$ | $1.06 \pm 0.01$ | $3.66 \pm 0.02$ |
| AHL | $97.36 \pm 0.23$ | $1.12 \pm 0.01$ | $3.76 \pm 0.02$ | $67.48 \pm 0.77$ | $1.09 \pm 0.01$ | $3.68 \pm 0.04$ |
| Goal-tracking | $98.51 \pm 0.45$ | $\mathbf{1.13 \pm 0.00}$ | $3.52 \pm 0.03$ | $72.60 \pm 0.97$ | $\mathbf{1.11 \pm 0.00}$ | $3.43 \pm 0.02$ |
| HWC-Loco-l | $99.95 \pm 0.02$ | $1.12 \pm 0.02$ | $\mathbf{3.43 \pm 0.04}$ | $78.92 \pm 0.45$ | $1.10 \pm 0.01$ | $\mathbf{3.39 \pm 0.05}$ |
| HWC-Loco | $\mathbf{100.0 \pm 0.00}$ | $1.12 \pm 0.02$ | $3.44 \pm 0.09$ | $\mathbf{84.34 \pm 0.43}$ | $1.07 \pm 0.03$ | $3.46 \pm 0.03$ |

## 5.2 ROBUSTNESS: STABLE CONTROL UNDER DISTURBANCES

**Robustness under Simulated Disturbances.** To demonstrate the robustness of HWC-Loco, we conduct disturbance tests in simulation. The robots are commanded to follow sampled velocities within the training distribution while navigating a uniformly mixed terrain comprising flat ground, obstacles, slopes, and stairs. Under these general locomotion settings, we design three types of disturbances as follows: 1) External force/torque disturbances, where random forces and torques (up to 200 N and 200 N·m) are applied to each robot link, either at a low frequency of 1 Hz or as constant perturbations lasting 0.5 seconds, following prior setups (Weng et al., 2023; Lee et al., 2020). 2) Impulse disturbances on the CoM, implemented by directly altering the robot's CoM velocity (Weng et al., 2023; Gu et al., 2024b). We simulate both low-impulse and high-impulse scenarios, with linear velocity perturbations of 0–1 m/s or 1–2 m/s, and angular velocity perturbations of 0–0.5 rad/s or 0.5–1.0 rad/s, respectively. 3) Payload disturbances, where an additional mass uniformly sampled from either 0–5 kg or 0–10 kg is added to upper-body links (Margolis & Agrawal, 2023; Radosavovic et al., 2024), along with internal perturbations from sensor and actuator noise (see Table 11 in Appendix A.6), to simulate unmodeled dynamics and varying mass distribution scenarios.

As shown in Table 2, HWC-Loco consistently achieves the best overall robustness across disturbance types, attaining the highest success rates while maintaining the lowest *ZMP Deviation*. Under constant disturbances, it reaches a success rate of **75.95%** with a ZMP Deviation of **6.61%**, evidencing effective resistance to strong, persistent forces. For low-impulse perturbations, all methods remain relatively strong, yet HWC-Loco still leads at **94.84%**. Under more challenging high-impulse pushes, the gap widens: HWC-Loco attains **81.27%** with **7.90%** ZMP, indicating that its recovery mechanism (e.g., rapid body lowering and arm swing) mitigates sudden impacts and expedites stability recovery. In payload scenarios, despite not being explicitly trained for added-mass conditions (e.g., +10 kg in the hands), HWC-Loco maintains the highest success rates with consistently low ZMP. This highlights the robustness of HWC-Loco to unseen disturbances. In summary, HWC-Loco couples high task success with consistently low ZMP deviation, yielding safer contact stability across disturbances.

**Robustness in Realistic Deployment.** We deploy HWC-Loco on a real humanoid robot and evaluate its performance under various external force disturbances. Figure 12 (see Appendix B.6) shows policy switching and associated changes in roll and pitch angles under continuous perturbations, including pushes, pulls, and kicks. Upon encountering disturbances, the robot promptly switches to the recovery policy, adjusting its posture and gait to regain stability. Importantly, the controller does not rely solely

Table 2: Robustness of locomotion under different disturbances.

| Policy | External Force/Torque (Low-freq.) | | Impulse on CoM (Low-imp.) | | Payload (Low) | |
|---|---|---|---|---|---|---|
| | Success Rate ↑ | ZMP ↓ | Success Rate ↑ | ZMP ↓ | Success Rate ↑ | ZMP ↓ |
| DreamWaQ | 85.92±1.64 | 5.94±0.08 | 85.24±0.37 | 6.88±0.08 | 67.63±3.10 | 18.35±0.21 |
| AHL | 87.15±2.86 | 6.42±0.10 | 85.87±1.47 | 7.53±0.11 | 79.29±0.65 | 17.50±0.19 |
| Goal-tracking | 90.00±2.20 | 7.64±0.07 | 88.90±0.60 | 8.24±0.11 | 78.34±1.23 | 10.41±0.18 |
| HWC-Loco-l | 92.69±0.56 | 5.59±0.08 | 92.79±1.75 | 6.16±0.10 | 84.11±2.66 | 10.37±0.11 |
| HWC-Loco | **95.88±0.37** | **5.53±0.07** | **94.84±0.54** | **6.13±0.09** | **87.43±0.92** | **9.33±0.12** |

| Policy | External Force/Torque (Constant) | | Impulse on CoM (High-imp.) | | Payload (High) | |
|---|---|---|---|---|---|---|
| | Success Rate ↑ | ZMP ↓ | Success Rate ↑ | ZMP ↓ | Success Rate ↑ | ZMP ↓ |
| DreamWaQ | 51.31±0.97 | 7.71±0.09 | 45.34±0.41 | 8.97±0.12 | 55.04±2.82 | 18.67±0.27 |
| AHL | 60.72±0.36 | 7.96±0.07 | 62.94±1.29 | 8.99±0.10 | 59.16±1.60 | 17.74±0.14 |
| Goal-tracking | 61.20±0.61 | 8.60±0.08 | 58.13±0.78 | 10.17±0.10 | 61.44±1.00 | 10.81±0.16 |
| HWC-Loco-l | 68.60±0.28 | 6.62±0.07 | 77.09±0.79 | 8.18±0.09 | 63.96±0.09 | 10.68±0.13 |
| HWC-Loco | **75.95±0.66** | **6.61±0.09** | **81.27±0.80** | **7.90±0.08** | **69.86±1.00** | **9.69±0.11** |

on recovery mode but dynamically switches between goal-tracking and recovery policies, thereby adapting the action distribution to environmental changes. Figure 13 (see Appendix B.6) further demonstrates the robot's robustness to disturbances in outdoor settings.

### 5.3 NATURALNESS: IMITATING HUMAN-LIKE MOVEMENTS

As shown in Figure 4, the robot remains standing at 0 m/s, and exhibits natural walking at 1.0 m/s, characterized by straight elbows and coordinated arm swings. At 2.5 m/s, it transitions to running, with longer strides and increasingly bent elbows, closely resembling human running. As the velocity increases, both stride length and step frequency adjust dynamically, mimicking typical human locomotion. In contrast, prior approaches (Gu et al., 2024b;c; Cui et al., 2024; Zhang et al., 2024b) often maintain a fixed step frequency regardless of the commanded velocity. Some even exhibit in-place stepping at 0 m/s (Zhang et al., 2024b; Cui et al., 2024), resulting in unnatural behavior for tasks requiring a steady stance, such as manipulation.

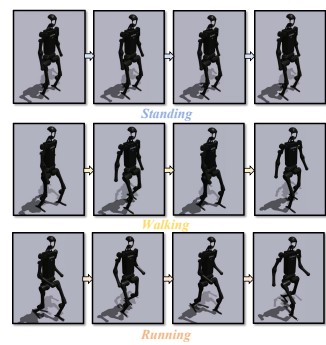

*Standing*

*Walking*

*Running*

Figure 4: Human-like behavior.

### 5.4 SCALABILITY: EXTENDING HWC-LOCO TO NEW TASKS

**Cross Embodiment Validation.** We deploy HWC-Loco on the Unitree G1 humanoid robot to evaluate its generalization capability under the same locomotion settings. As shown in Table 3, G1 outperforms H1 across all metrics. A detailed analysis is provided in Appendix B.7.

Table 3: Comparison of Embodiments

| | Success Rate ↑ | Goal Tracking ↑ | Human-like ↓ |
|---|---|---|---|
| Unitree H1 | 97.13 ± 0.43 | 1.10 ± 0.00 | 3.18 ± 0.01 |
| Unitree G1 | 98.14 ± 0.35 | 1.12 ± 0.01 | 3.11 ± 0.03 |

Table 4: Motion Tracking under Disturbances

| | Punching ↑ | Dancing ↑ | Expressive Walking ↑ |
|---|---|---|---|
| HWC-Loco | 94.01 ± 0.49 | 86.44 ± 0.60 | 94.53 ± 0.23 |
| Motion-tracking | 90.85 ± 0.74 | 81.64 ± 0.64 | 89.63 ± 0.27 |

**Expressive Motion Tracking.** Following ExBody (Cheng et al., 2024; Ji et al., 2024), we set expressive motion as HWC-Loco's goal and train the model using the same locomotion pipeline. A domain-randomized motion-tracking policy serves as the baseline. We evaluate both on three representative motions under impulse disturbances, and report success rates in Table 4.

## 6 LIMITATION

Our approach has three limitations. First, policy switching is handled by a discrete module while low-level controllers are frozen. Jointly optimizing the hierarchy could yield smoother transitions. Second, the humanoid robot in real-world deployment has only 19 DOF, which limits whole-body coordination and constrains the expression of complex recovery behaviors. Third, the recovery policy is trained in simulation and may not capture the full diversity of real-world disturbances. Incorporating more realistic or adversarial perturbations could further improve robustness.

## 7 CONCLUSION

We introduce HWC-Loco, a hierarchical control framework for humanoid robots that incorporates an embedded safety recovery mechanism. This framework has been validated across various tasks and

scenarios, demonstrating exceptional effectiveness, robustness, naturalness, and scalability. Notably, the safety mechanism in HWC-Loco extends beyond locomotion, enabling reliable performance in complex tasks through a dynamic task-safety balance. This ensures robust operation in real-world deployments, positioning HWC-Loco as a foundational solution for safety-critical applications such as industrial automation and assistive robotics. A promising direction for future research is integrating HWC-Loco with loco-manipulation skills, enabling safety-critical control across a broader range of tasks involving different objects.

## ETHICS STATEMENT

This research concerns robotic control and does not involve human participants, personally identifiable data, or sensitive attributes. It presents no harmful insights, dual-use risks, or discrimination concerns. We disclose no conflicts of interest or external sponsorship; accordingly, we do not foresee ethical issues.

## REPRODUCIBILITY STATEMENT

We provide the implementation of HWC-Loco in the supplementary material, and the hyperparameter settings are detailed in Appendix A, enabling faithful replication of our results.

## ACKNOWLEDGMENTS

This work is supported in part by Shenzhen Science and Technology Program under grant KJZD20240903104008012, Shenzhen Science and Technology Program under grant ZDCY20250901113000001, CUHK-CUHK(SZ)-GDSTC Joint Collaboration Fund No. 2025A0505000053, GuangDong Key Laboratory of Big Data Computing (2021B1212040002) and Guangdong Provincial Key Laboratory of Mathematical Foundations for Artificial Intelligence (2023B1212010001).

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

# A    ADDTIONAL DETAILS OF HWC-LOCO

## A.1    MOTION RETARGETING

Our motion data is derived from the CMU Mocap dataset cmu, which exclusively encompasses human locomotion data, featuring a diverse array of walking, jogging, and running styles. In aggregate, we have loaded 318 motion sequences, totaling 3729.18 seconds in duration. Our motion retargeting methodology closely follows the approach outlined in Cheng et al. (2024). We have meticulously aligned the humanoid robot's skeleton with that of humans, incorporating adjustments such as rotation, translation, and scaling. Additionally, we map human joints onto the humanoid robot's structure. Specifically, the human motion data is adapted to the humanoid robot's framework by mapping local information. Both the Unitree H1 and G1 robots possess shoulder and hip joints that are functionally equivalent to spherical joints. During the retargeting process, the three hip and shoulder joints are treated as a single spherical joint, corresponding to the human's spherical joint. For one-dimensional joints such as the elbow and torso, the rotation angle is projected onto the corresponding rotation axis of the joint angles.

Table 5: Double-DQN Parameters

| Parameter | Value |
|---|---|
| Batch Size | 128 |
| Learning Rate | 1e-4 |
| Gamma | 0.99 |
| Max Grad Norm | 1.0 |
| Replay Buffer Capacity | 2000000 |
| Switch Penalty Coefficient | 0.001 |
| Target Update Frequency | 50 |
| Optimizer | Adam |
| Loss Function | MSELoss |
| Initial Epsilon | 0.1 |
| Minimum Epsilon | 0.001 |
| Epsilon Decay Rate | 0.999 |

Table 6: DQN Network Structure

| Layer | Details |
|---|---|
| Layer 1 | [input_dim, 256] |
| Layer 2 | Activation: ReLU |
| Layer 3 | Regularization: Dropout |
| Layer 4 | [256, 128] |
| Layer 5 | [128, 64] |
| Layer 6 | Activation: ReLU |
| Layer 7 | [64, output_dim] |

## A.2    IMPLEMENTATION DETAILS

**Proprioception:** For the Unitree H1, proprioception $o_t^p \in \mathbb{R}^{65}$, which denotes the internal state of the robot, including the base angular velocity $w_t \in \mathbb{R}^3$, base roll $r_t \in \mathbb{R}^1$, base pitch $p_t \in \mathbb{R}^1$, degrees of freedom (DOF) positions $q_t \in \mathbb{R}^{19}$, DOF velocities $\dot{q}_t \in \mathbb{R}^{19}$, previous actions $a_{t-1} \in \mathbb{R}^{19}$, and projected gravity $g_t \in \mathbb{R}^3$. The projected gravity refers to the component of gravity expressed in the robot's local coordinate system. For the Unitree G1, proprioception $o_t^p \in \mathbb{R}^{77}$, includes base angular velocity $w_t \in \mathbb{R}^3$, base roll $r_t \in \mathbb{R}^1$, base pitch $p_t \in \mathbb{R}^1$, DOF positions $q_t \in \mathbb{R}^{23}$, DOF velocities $\dot{q}_t \in \mathbb{R}^{23}$, previous actions $a_{t-1} \in \mathbb{R}^{23}$, and projected gravity $g_t \in \mathbb{R}^3$. As with the H1, the projected gravity denotes the component of gravity in the robot's local coordinate system.

**Action Space:** The policy outputs continuous actions $\mathbf{a}_t \in \mathbb{R}^n$, which are utilized as target positions for a PD controller to compute joint torques. The actions correspond to the robot's actuated degrees of freedom: specifically, $\mathbf{a}_t \in \mathbb{R}^{19}$ (19 DoF) for the Unitree H1 and $\mathbf{a}_t \in \mathbb{R}^{23}$ (23 DoF) for the Unitree G1.

**Terrain details:** Terrains in the training environment simulated by Isaac Gym Makoviychuk et al. (2021) include flats, obstacles, slopes, and stairs. An example of these terrains is shown in Figure 5.

**Low-level Policy Training:** For all the low-level policies training, the commands are sampled from ranges: $v_x^c \sim [-0.6, 2.5]$, $v_y^c \sim [-0.6, 0.6]$ and $w_z^c \sim [-1.0, 1.0]$. During the initial training of the humanoid robot on uneven terrain, the robot often remains stationary due to the high difficulty of navigating complex surfaces. To address this, we introduce a terrain curriculum method Makoviychuk et al. (2021). The training terrain consists of various types, including flat planes, rough surfaces, steps, and slopes. As the robot achieves a goal-tracking performance of 70% of the commanded

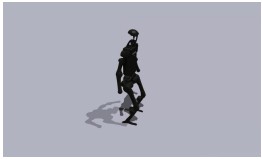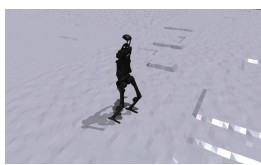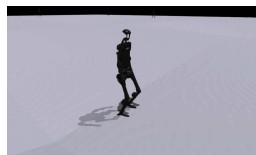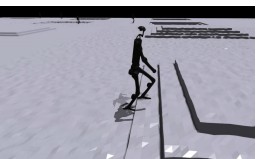

Figure 5: Visualization of different terrains. From left to right, the terrains are flats, obstacles, slopes, and stairs

velocity, the terrain difficulty increases. Conversely, if the goal-tracking performance drops below 40% of the commanded velocity, the terrain becomes easier.

**Learning for Policy Transition:** High-level planning policy is utilized to switch between low-level policies, ensuring compliance with robustness constraints. For the High-level policy, the input is the same set of observations as used by the low-level policies, with the output being a two-dimensional Q-value. During training, two trained low-level policies are loaded and rolled out to generate training data for optimizing the high-level policy. We train this high-level policy in the simulation environment mentioned detailed in Section 4.2, focusing on locomotion goals as the primary task. The objective is to enable the robot to track goal commands across a variety of terrains. The details of the training process is in Algorithm 1

---

**Algorithm 1** High-level Policy Training

---

**Input:** Environment $env$, Selector network $S$, Target network $S_{\text{target}}$, Goal-tracking policy $L$, Recovery policy $R$
Initialize: Learning rate $\alpha$, Discount factor $\gamma$, Exploration rate $\epsilon$, Replay buffer $B$, Target update frequency $f_{\text{target}}$, Switch penalty $\lambda_{\text{switch}}$, Exploration decay $\epsilon_{\text{decay}}$, Minimum exploration $\epsilon_{\text{min}}$
$S_{\text{target}} \leftarrow S$ {Initialize target network}
**repeat**
    Observe initial state $s$ from $env$
    Reset cumulative reward $R_{\text{total}}$, step count $n$, previous action $a_{\text{prev}} \leftarrow$ None
    **while** not done **do**
        Compute $Q(s, \cdot) \leftarrow S(s)$, select action $a$ with $\epsilon$-greedy
        Compute switch penalty: $r_{\text{penalty}} \leftarrow \lambda_{\text{switch}} \cdot \mathbf{1}[a \neq a_{\text{prev}}]$
        Execute $a$, observe reward $r$, next state $s'$, and done $d$
        Store $(s, a, r + r_{\text{penalty}}, s', d)$ in $B$
        Update $s \leftarrow s'$, $a_{\text{prev}} \leftarrow a$
        $R_{\text{total}} \leftarrow R_{\text{total}} + r$, $n \leftarrow n + 1$
        **if** $B$.size() $\geq$ batch size **then**
            Sample $(s, a, r, s', d)$ from $B$
            $a' \leftarrow \arg\max_{a'} S(s', a')$ {Online network selects action}
            $Q^{\text{target}}(s', a') \leftarrow S_{\text{target}}(s', a')$ {Target network evaluates}
            Compute target: $Q^{\text{target}}(s, a) \leftarrow r + \gamma(1 - d)Q^{\text{target}}(s', a')$
            Update $S$ by minimizing: $\mathcal{L} = (S(s, a) - Q^{\text{target}}(s, a))^2$
        **end if**
        **if** $n \bmod f_{\text{target}} = 0$ **then**
            $S_{\text{target}} \leftarrow S$ {Sync target network}
        **end if**
    **end while**
    Decay exploration rate: $\epsilon \leftarrow \max(\epsilon \cdot \epsilon_{\text{decay}}, \epsilon_{\text{min}})$
**until** convergence criteria met
**Output:** Trained Selector Network $S$

---

**Reward Details:** To better facilitate efficient humanoid locomotion, our task reward is defined as Gu et al. (2024b;c):

$$r_{\mathfrak{T}} = \alpha_1 \exp\left(-\frac{\|v_{xy}^c - v_{xy}\|_2^2}{\sigma_{lin}^2}\right) + \alpha_2 \exp\left(-\frac{\|w_z^c - w_z\|_2^2}{\sigma_{ang}^2}\right)$$

where 1) $v_{xy}^c$ denotes the commanded linear velocity along the $x$ and $y$ axes, while $w_z^c$ is the commanded yaw velocity. 2) $v_{xy}$ and $w_z$ represent the corresponding base velocities of the humanoid robot. 3) $\alpha_1$ and $\alpha_2$ indicate the hyperparameters that are utilized to adjust the importance of the different velocity terms. 4) $\sigma_{lin}^2$ and $\sigma_{ang}^2$ control the precision of the expected command tracking. Smaller values of $\sigma_{lin}^2$ and $\sigma_{ang}^2$ enhance the precision of command tracking for humanoid robots. However, they may hinder reward acquisition during the initial stages of training.

The recovery policy shares the same task reward formulation as the goal-tracking policy. However, it uses larger values of the $\sigma_{lin}^2$ and $\sigma_{ang}^2$, which implies a greater tolerance for velocity tracking errors. As a result, this reward term serves as a back-tracking reward for the safety recovery mechanism, encouraging it to return to a stable goal-tracking state. To further promote stable posture restoration and enable smooth transitions back to the goal-tracking policy, we introduce an additional stand reward, defined as:

$$r_{stand} = \|\mathbf{q}_t - \mathbf{q}_{default}\|_2^2.$$

where $\mathbf{q}_{default}$ represents a default standing pose.

For safety realistic deployment, we add some energy and safety reward for the low-level policies training. These rewards are shown in Table.

Table 7: Safety & Energy Reward

| Term | Expression | Weight |
|------|------------|--------|
| DoF position limits | $\mathbf{1}(\mathbf{q}_t \notin [\mathbf{q}_{min}, \mathbf{q}_{max}])$ | $-1e^2$ |
| DoF acceleration | $\|\ddot{\mathbf{q}}_t\|_2^2$ | $-1e^{-7}$ |
| DoF velocity | $\|\dot{\mathbf{q}}_t\|_2^2$ | $-5e^{-4}$ |
| Action rate | $\|\mathbf{a}_t - \mathbf{a}_{t-1}\|_2^2$ | $-2e^{-3}$ |
| Torque | $\|\tau_t\|$ | $-1e^{-5}$ |
| Collision | $\sum_{i \in contact} \mathbf{1}\{\|F_i\| > 1.0\}$ | $-10$ |

## A.3 PPO

In order to enhance the understanding of historical observation information, we have made some improvements to the Actor-network. We first process this information using an encoder network to preliminarily extract the features of the observation values at each time step, and then use a merger network to integrate the observation values at different time steps. Finally, we obtain fixed-dimensional feature information, which is then input into the Actor backbone network together with other information.

Table 8: PPO hyperparameters

| Hyperparameter | Value |
|---|---|
| Actor Layer | [512, 256, 128] |
| Critic Layer | [512, 256, 128] |
| Activation Function | ELU |
| Discount Factor | 0.99 |
| GAE Parameter | 0.95 |
| Epochs per Rollout | 5 |
| Minibatch | 4 |
| Entropy Coefficient | 0.005 |
| Value Loss Coefficient ($\alpha_1$) | 1.0 |
| Clip Range | 0.2 |
| Learning Rate | 2.e-4 |
| Environment Steps per Rollout | 24 |

## A.4 VAE

The VAE Network consists of an encoder and a decoder. The encoder takes historical observations as input and outputs latent representation and estimation of privileged information. The decoder takes latent as input and outputs the reconstruction of the next observation. The structure of VAE is shown in Figure 6

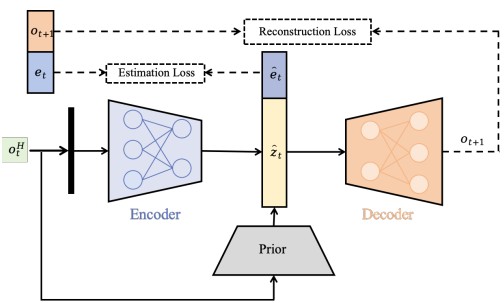

Figure 6: VAE Structure

Table 9: VAE hyperparameters

| Hyperparameter | Value |
|---|---|
| Encoder Layer | [256, 128] |
| Decoder Layer | [256, 128, 64] |
| Activation Function | ELU |
| KL Coefficient | 1.0 |
| Learning Rate | 1.e-4 |

## A.5 DISCRIMINATOR

The input to the discriminator $s^d$ is defined as $[q, v, w, r, p, k]$, where $k$ represents the local positions of key points. Specifically, we define 12 key points, including the hips, shoulders, hands, ankles, knees, and elbows. For the Unitree H1, the total input dimension is 63, while for the Unitree G1, it is 67.

The discriminator's hyperparameters are shown in Table. 10.

Table 10: Discriminator hyperparameters

| Hyperparameter | Value |
|---|---|
| Discriminator Hidden Layer Dim | [1024,512] |
| Replay Buffer Size | 500000 |
| Expert Buffer Size | 200000 |
| Expert Fetch Size | 512 |
| Learning Batch Size | 4096 |
| Learning Rate | 2e-5 |
| Gradient Penalty Coefficient | 1.0 |
| lambda | 1.0 |

### A.6 EXTREME INITIALIZATION

To simulate extreme scenarios that may occur in real-world deployment, we identify three primary causes of such cases: external disturbances, hardware malfunctions, and malicious commands. Additionally, we retain domain randomization to enhance the generalization of physical dynamics within the simulation environment. The specific details are outlined as follows:

To enhance robustness and generalization in simulation, we simulate both extreme deployment scenarios and apply domain randomization. The extreme scenarios include external disturbances, sensor and actuator failures, and adversarial commands, while domain randomization targets physical variability in the environment.

1) **External disturbances:** We apply external disturbances to every component of the robot. The external forces and torques are applied along the x, y, and z axes, with each direction sampled from the range of -200 N to 200 N for forces and -200 Nm to 200 Nm for torques. These disturbance forces and torques are updated at a frequency of 1 Hz.

2) **Hardward magnification :** Random, high-intensity noise is introduced to the proprioceptive information and PD gains. The noise added to the proprioceptive data simulates sensor errors that may occur during real-world deployment. The noise applied to the PD gains simulates fluctuations in motor strength, reflecting potential issues in the robot's actuators. The parameters for this noise are detailed in Table 11.

3) **Malicious velocity:** Velocity commands are resampled from the command range, with updates occurring at a frequency of 5 Hz. Notably, this resampling mechanism is activated once the terrain curriculum exceeds half of its total progression.

4) **Domain randomization (for environment variability):** Domain randomization is applied to all the training environments. This technique introduces variability in the environment's parameters, such as friction and gravity. The specifics of the domain randomization are provided in Table 12.

To improve the generalization of recovery ability, we utilize random initialization with extreme case initialization. The parameters are shown in Table 13

Table 11: Intense Noise Parameters

| Parameter | Range | Operator |
|---|---|---|
| DOF Position Noise | 0.04 | additive |
| DOF Velocity Noise | 0.30 | additive |
| Angular Velocity Noise | 1.00 | additive |
| IMU Noise | 0.40 | additive |
| Gravity Noise | 0.10 | additive |
| PD Gains | 0.20 | additive |

Table 12: Domain Randomization Parameters

| Parameter | Range | Operator |
|---|---|---|
| Friction | [0.6, 2.0] | additive |
| Base Mass | [-1.0, 5.0] | additive |
| Base CoM | [-0.07, 0.07] | additive |
| Motor Strength | [0.8, 1.2] | scaling |
| Action Delay | [0, 5] | - |
| Link Mass | [0.7, 1.3] | scaling |
| Gravity | [-0.1, 0.1] | additive |

Table 13: Random Initialization Parameters

| Parameter | Range |
|---|---|
| Linear Velocity (x) | $[-1.0, 2.5]$ |
| Linear Velocity (y) | $[-1.0, 1.0]$ |
| Linear Velocity (z) | $[-0.4, 0.4]$ |
| Angular Velocity | $[-1.0, 1.0]$ |
| DOF Position | $[0.2 \times \text{default}, 1.8 \times \text{default}]$ |
| DOF Velocity | $[-0.2, 0.2]$ |
| Pitch Angle | $[-0.25, 0.25]$ |
| Yaw Angle | $[-0.4, 0.4]$ |
| Position | $[-2, 2]$ |

**Extreme-Case Analysis:** We record nearly 60000 normal state data with randomly sampled commands in diverse terrain settings in 7. We record nearly 10000 extreme cases in the $\mathfrak{P}_\alpha^L$. Some of

the terminal states are visualized in Fig. 8. For the extreme states, the distribution of velocity is close to the command ranges. Roll pitch and their velocity are in a common range, which is near zeros. For the normal states, the distribution of linear velocity overlaps with the command velocity range, with a dense cluster around the high-velocity value of 2 m/s. Regarding yaw velocity, most values fall within the commanded range. However, there are instances of high speed, with some even approaching 10 m/s. This may be due to the robot's extreme behaviors when encountering unseen situations influenced by external forces and random noise. Regarding the roll-pitch velocity, the pitch velocity shows a higher degree of fluctuation. Similarly, in the roll pitch distribution, for pitch, there is a concentrated distribution at a pitch of 1 m/s, which is one of the terminal conditions. These two figures suggest that the robot is tripped by obstacles during its motion and thus terminates, for instance, when it hits steps or gravel during high-speed motion.

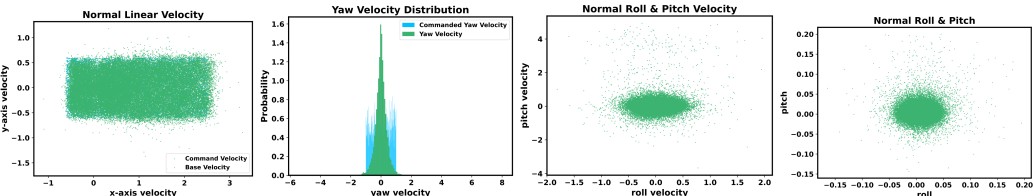

Figure 7: Normal State: Policy rollout's state distribution in the original learning environment

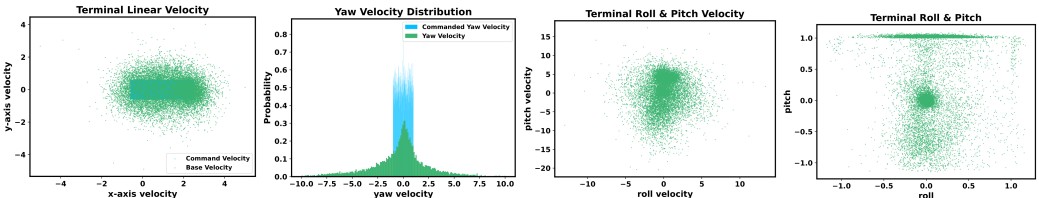

Figure 8: Extreme State: Policy's state distribution in the extreme cases

### A.7 BASELINE REPRODUCIBILITY

We adapt baseline implementations (DreamWaQ Nahrendra et al. (2023), AHL Cui et al. (2024)) as faithfully as possible while aligning embodiment and simulator specifics for fair comparison and reproducibility.

**Common adaptation principles.** 1) Proprioceptive inputs, action space, PD gains, control frequency, and domain–randomization ranges are aligned with HWC-Loco. 2) Physics time step, contact parameters, and terrains are matched to our Isaac Gym setup. 3) Training/validation splits, evaluation episodes, and disturbance samplers are shared across methods.

**DreamWaQ.** We retain the original reward structure and network design, with minimal adaptations to account for humanoid morphology and dynamics: 1) reward *scales* are slightly adjusted (see Table 14); 2) a VAE-based estimator predicts velocity and base height from short observation histories; 3) following the original setting, the policy consumes a single observation frame (no temporal stacking). All other simulation and control parameters follow the common principles above.

**AHL.** AHL adopts a gait-centric objective with phase tracking, foot slip, and air time. We reuse the same estimator as DreamWaQ, while the policy ingests short histories of observations (temporal stacking). We employ a two-stage curriculum: 1) train on simple terrains with strong gait-shaping terms; 2) once the success rate exceeds 80%, remove the gait-shaping bonus and increase terrain difficulty and command ranges.

**Reproducibility notes.** We use shared disturbance samplers and evaluation protocols for all baselines; policy observation formats, actuator models, PD targets, and randomization ranges are docu-

Table 14: DreamWaQ reward terms and weights.

| Reward term | Equation | Weight |
|---|---|---|
| Linear vel. tracking | $\exp\left\{-5\,\|v_{xy}^{\text{cmd}} - v_{xy}\|^2\right\}$ | 1.0 |
| Angular vel. tracking | $\exp\left\{-5\,(\omega_{\text{yaw}}^{\text{cmd}} - \omega_{\text{yaw}})^2\right\}$ | 0.8 |
| Velocity mismatch | $\exp\left\{-10\,(-v_z)^2\right\} + \exp\left\{-5\,\|\omega_{\text{roll,pitch}}\|^2\right\}$ | 0.5 |
| Orientation | $\|g\|^2$ | 1.0 |
| Default joint position | $\exp\left\{-2\,\|\theta - \theta_{\text{zero}}\|^2\right\}$ | 0.5 |
| Body height | $\left(h^{\text{des}} - 1.0\right)^2$ | $-1.0$ |
| Root accelerations | $\exp\left\{-\|\ddot{\theta}_{\text{root}}\|^3\right\}$ | 0.2 |
| Joint accelerations | $\|\ddot{\theta}\|^2$ | $-1\times10^{-7}$ |
| Joint velocity | $\|\dot{\theta}\|^2$ | $-5\times10^{-4}$ |
| Joint energy | $\|\tau\|^2$ | $-1\times10^{-5}$ |
| Action smoothness | $\|a_t - a_{t-1}\|^2 + \|a_t - 2a_{t-1} + a_{t-2}\|^2$ | $-0.002$ |
| Feet clearance | $\|p_f^{\text{des}}(t) - p_f(t)\|^2 \cdot \left(1 - I_d(t)\right)$ | $-0.01$ |
| Feet & Knee distance | $\frac{1}{2}\left[\exp\left(-100\,|\min(d_f - d_{\min}, 0)|\right) + \exp\left(-100\,\max(d_f - d_{\max}, 0)\right)\right]$ | 0.2 |

Table 15: AHL gait-centric reward terms (additional to the shared terms in Table 14).

| Reward term | Equation | Weight |
|---|---|---|
| Gait phase tracking | $\left(1 - I_d(t)\right) \cdot I_c(t)$ | $-0.001$ |
| Feet air time | $T_{\text{air}} \cdot I_c(t)$ | 10.0 |
| Feet slip | $\omega \cdot I_c(t)$ | $-0.01$ |
| Joint position tracking | $\exp\left\{-2\,\|\theta - \theta_{\text{target}}\|^2\right\}$ | 1.0 |
| Other terms | Linear/Angular velocity, Height, Feet clearance/distance, etc. | same as Table 14 |

mented in AppendixA.2 and AppendixA.6 Code will be released with configuration files specifying seeds, observation keys, disturbance schedules, and reward weights corresponding to Tables 14–15.

## A.8 COMPUTATIONAL COST

All policies are trained on a single NVIDIA RTX 4090 GPU with 4,096 parallel environments. Our framework involves training three policies. The goal-tracking policy converges in roughly 10,000 iterations (about 10 hours), while the recovery policy requires around 8,000 iterations (about 8 hours). The high-level policy converges fastest, needing approximately 6,000 iterations (under 6 hours). For comparison, DreamWaQ and AHL require about 8 and 9 hours of training, respectively, under the same simulation settings.

## A.9 DEPLOYMENT DETAILS

We utilize the Unitree H1 humanoid robot as our deployment platform. This full-sized humanoid robot weighs 47 kg, stands approximately 1.8 meters tall, and features 19 degrees of freedom. The control frequency is set to 100 Hz for both simulation and real-world deployment. The PD gains, characterized by stiffness and damping values, used in the real-world deployment are detailed in Table 16.

For smoother policy transitions, we apply a Butterworth low-pass filter (cutoff frequency: 5 Hz), following He et al. (2024b), during real-world deployment to attenuate switching-induced transient oscillations. During deployment, inference operates through a hierarchical structure involving both high-level and low-level policies. To meet the strict latency and resource constraints of onboard systems, the policies are optimized for lightweight designs and exported JIT format. The high-level policy model size is below 2MB, and the low-level policies remain under 3MB. Those designs ensures that real-time inference at 100 Hz is feasible on the onboard computer of the Unitree H1 robot.

Table 16: Stiffness, Damping and Torque Limit

| Joint Names | Stiffness [N*m/rad] | Damping [N*m*s/rad] | Torque Limit (Nm) |
|---|---|---|---|
| hip yaw | 200 | 5 | 170 |
| hip roll | 200 | 5 | 170 |
| hip pitch | 200 | 5 | 170 |
| knee | 300 | 6 | 255 |
| ankle | 40 | 2 | 34 |
| torso | 200 | 5 | 170 |
| shoulder | 30 | 1 | 34 |
| elbow | 30 | 1 | 18 |

# B  EXPERIMENT

## B.1  SALIENCY ANALYSIS OF OBSERVATIONS

We adopt an integrated gradients method similar to Wang et al. (2024) to assess the importance of different parts of the observation to the policy, as shown in Figure 9.

For the recovery policy, the ZMP encoding proves to be the most critical factor, followed by the estimated velocity. The robot's current and historical proprioceptive states also contribute substantially to the policy's decision-making process. In contrast, the environmental latent representation exhibits only a marginal effect. This hierarchy of importance reflects the necessity of strictly adhering to ZMP constraints to maintain dynamic stability and ensure the safety of the robot during recovery.

For the high-level policy, the ZMP encoding remains the most influential component, indicating that it provides critical information for decision-making and substantially enhances the policy's awareness of the robot's stability. This observation implies that the high-level policy prioritizes stability-relevant features over task-irrelevant ones, which is advantageous for promoting generalization across diverse scenarios. Moreover, the environmental latent variable exhibits greater importance compared to that in the recovery policy, suggesting that high-level decisions are more sensitive to environmental factors such as terrain variations.

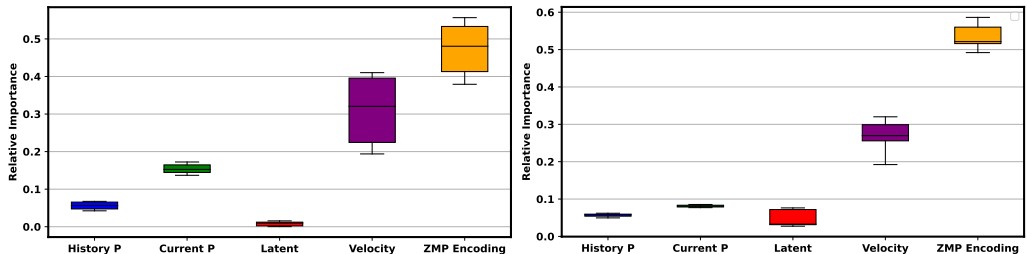

Figure 9: Importance of Observations

## B.2  SENSITIVITY TO HYPERPARAMETERS

**Sensitivity to $\alpha$.** To quantify the sensitivity to the mismatch scale $\alpha$, we scale all disturbance and parameter ranges by factors $\{0.5, 1.0, 2.0\}$, where the original setting corresponds to $\alpha = 1.0$. As shown in Table 17, smaller $\alpha$ improves nominal tracking but reduces robustness, while larger $\alpha$ yields more conservative yet more robust behavior.

Table 17: Sensitivity of HWC-Loco to the mismatch scale $\alpha$.

| Policy | Success Rate ↑ | Goal-tracking ↑ | Human-like ↓ |
|---|---|---|---|
| HWC-Loco (0.5) | $97.88 \pm 0.26$ | $\mathbf{1.14 \pm 0.24}$ | $\mathbf{3.10 \pm 0.11}$ |
| HWC-Loco (1.0) | $98.14 \pm 0.35$ | $1.12 \pm 0.01$ | $3.11 \pm 0.03$ |
| HWC-Loco (2.0) | $\mathbf{98.20 \pm 0.33}$ | $1.10 \pm 0.05$ | $3.12 \pm 0.14$ |

**Sensitivity to $\lambda$.** To quantify sensitivity to imitation weight $\lambda$. The imitation weight $\lambda$ controls the strength of the Wasserstein imitation objective for $\pi_1$. We start from the nominal value used in the main experiments and perform a small sweep around it by scaling $\lambda$ by factors $\{0.5, 1.0, 2.0\}$. As summarized in Table 18, smaller $\lambda$ slightly improves raw command tracking but produces less human-like, stiffer motion. Larger $\lambda$ yields smoother and more human-like gaits, at the cost of a modest increase in tracking error. When combined with the *same* $\pi_2$ and $\pi_0$, robustness under disturbances changes only marginally across this range, since recovery capability is primarily governed by $\pi_2$ and the switching behavior of $\pi_0$. Overall, HWC-Loco is not overly sensitive to the exact choice of $\lambda$ within this range.

Table 18: Sensitivity of HWC-Loco to the imitation weight $\lambda$.

| Policy | Success Rate ↑ | Goal-tracking ↑ | Human-like ↓ |
|---|---|---|---|
| HWC-Loco (0.5) | $\mathbf{98.21 \pm 0.14}$ | $1.12 \pm 0.01$ | $3.23 \pm 0.03$ |
| HWC-Loco (1.0) | $98.14 \pm 0.35$ | $\mathbf{1.12 \pm 0.01}$ | $3.11 \pm 0.03$ |
| HWC-Loco (2.0) | $98.08 \pm 0.26$ | $1.12 \pm 0.02$ | $\mathbf{3.08 \pm 0.03}$ |

### B.3 Comparison to Strong Domain Randomization Policies

To further clarify the benefit of our structural safety-recovery hierarchy, we compare HWC-Loco against strong domain-randomized (DR) baselines under *extreme* disturbance settings. In addition to our standard DR baselines, we introduce a strong non-hierarchical baseline, denoted *Large-DR-Hist*, which uses a single history-aware policy trained with large-scale domain randomization and regularization rewards, and is evaluated at randomization scales 1, 2, and 4. This baseline shares the same randomized dynamics and disturbance distributions as our safety-recovery policy. As summarized in Table 19, under comparable DR configurations HWC-Loco achieves clearly superior performance in most scenarios, with substantially fewer catastrophic failures and more successful recoveries, especially under constant external forces and high-impulse disturbances. These results highlight the value of the proposed safety-critical hierarchy on top of DR.

Table 19: Comparison with strong DR baselines under extreme disturbances.

| Policy | Low-freq. ↑ | Constant ↑ | Low-imp. ↑ | High-imp. ↑ | Low Payload ↑ | High Payload ↑ |
|---|---|---|---|---|---|---|
| Large-DR-Hist (1.0) | $87.15 \pm 0.23$ | $61.72 \pm 0.36$ | $85.45 \pm 0.87$ | $62.38 \pm 1.03$ | $79.53 \pm 0.76$ | $60.23 \pm 0.61$ |
| Large-DR-Hist (2.0) | $88.45 \pm 0.56$ | $65.32 \pm 0.28$ | $88.47 \pm 1.75$ | $68.36 \pm 0.79$ | $82.32 \pm 0.87$ | $65.21 \pm 0.57$ |
| Large-DR-Hist (4.0) | $90.35 \pm 0.43$ | $70.53 \pm 0.42$ | $90.67 \pm 1.31$ | $71.36 \pm 0.89$ | $85.77 \pm 0.93$ | $\mathbf{70.36 \pm 0.69}$ |
| **HWC-Loco** | $\mathbf{95.88 \pm 0.37}$ | $\mathbf{75.95 \pm 0.66}$ | $\mathbf{94.84 \pm 0.54}$ | $\mathbf{81.27 \pm 0.80}$ | $\mathbf{87.43 \pm 0.92}$ | $69.86 \pm 1.00$ |

### B.4 Robustness to VAE Estimation Noise

To quantify the quality of the VAE estimates, we report the mean squared error of the reconstructed velocity and ZMP-related features under different disturbance scenarios in Table 20. The errors remain small across all settings, and only increase moderately under strong impulses or heavy payloads.

We further evaluate the robustness of $\pi_0$ to additional synthetic noise by injecting Gaussian noise with standard deviation $\sigma$ into the VAE-estimated features at test time. As shown in Table 21, for all *realistic* noise levels ($\sigma \le 1.0$), the success rate remains high ($> 76\%$) and the number of switches (*Flip Count*) increases only mildly. Performance degrades substantially only under extreme noise

Table 20: VAE estimation accuracy across disturbances.

| Scenario | Velocity ↓ | ZMP Features ↓ |
|---|---|---|
| Normal | $0.0250 \pm 0.0008$ | $\mathbf{0.0491 \pm 0.0005}$ |
| Low-Frequency | $\mathbf{0.0242 \pm 0.0009}$ | $0.0493 \pm 0.0008$ |
| Constant | $0.0294 \pm 0.0009$ | $0.0507 \pm 0.0009$ |
| Low-Impulse | $0.0408 \pm 0.0013$ | $0.0518 \pm 0.0016$ |
| High-Impulse | $0.0719 \pm 0.0014$ | $0.0531 \pm 0.0014$ |
| Low Payload | $0.0497 \pm 0.0012$ | $0.0577 \pm 0.0012$ |
| High Payload | $0.0510 \pm 0.0013$ | $0.0751 \pm 0.0018$ |

($\sigma = 2.0$), which is well beyond the observed VAE error. These results indicate that $\pi_0$ is robust to realistic estimation noise and does not exhibit significant chattering.

Table 21: Robustness of the high-level policy $\pi_0$ to injected VAE estimation noise.

| Noise Level | Success Rate (%) ↑ | Flip Count ↓ |
|---|---|---|
| $\sigma = 0.0$ | $\mathbf{78.65 \pm 0.53}$ | 80 |
| $\sigma = 0.1$ | $77.77 \pm 0.62$ | 81 |
| $\sigma = 0.2$ | $77.36 \pm 0.61$ | 82 |
| $\sigma = 0.5$ | $77.33 \pm 0.64$ | 84 |
| $\sigma = 1.0$ | $76.99 \pm 0.83$ | 96 |
| $\sigma = 2.0$ | $21.81 \pm 3.41$ | $\mathbf{126}$ |

## B.5 ABLATION ANALYSIS

**Switching Strategies Ablation.** We compare our learned high-level planner with a fixed-threshold, ZMP-based heuristic that triggers recovery whenever the estimated ZMP exceeds a margin (0.2 or 0.4). Although intuitive, this heuristic relies on a VAE-based ZMP estimate and is thus noise-sensitive, often over-triggering recovery on otherwise recoverable perturbations. As summarized in Tables 22, 23, 24, and 25, the fixed-0.2 rule attains success close to ours but incurs very high switch counts and degraded tracking, while fixed-0.4 reduces switching and improves tracking at the cost of success. In contrast, our learned planner exploits temporal context and jointly optimizes success, tracking, and switching: HWC-Loco achieves the best overall trade-off (highest success with competitive switching), whereas HWC-Loco-l minimizes switching and attains the best tracking.

Table 22: Low-Frequency Disturbance

| Method | Success Rate ↑ | Goal Tracking ↑ | Switch Count |
|---|---|---|---|
| HWC-Loco-fixed-0.2 | $94.83 \pm 0.31$ | $0.93 \pm 0.01$ | 461 |
| HWC-Loco-fixed-0.4 | $91.54 \pm 0.32$ | $1.00 \pm 0.01$ | 190 |
| HWC-Loco-l | $92.69 \pm 0.56$ | $\mathbf{1.04 \pm 0.01}$ | 21 |
| HWC-Loco | $\mathbf{95.88 \pm 0.37}$ | $1.02 \pm 0.01$ | 183 |

Table 23: Constant Disturbance

| Method | Success Rate ↑ | Goal Tracking ↑ | Switch Count |
|---|---|---|---|
| HWC-Loco-fixed-0.2 | $70.71 \pm 0.63$ | $0.86 \pm 0.01$ | 459 |
| HWC-Loco-fixed-0.4 | $64.86 \pm 0.78$ | $0.95 \pm 0.01$ | 206 |
| HWC-Loco-l | $68.60 \pm 0.28$ | $\mathbf{0.99 \pm 0.01}$ | 45 |
| HWC-Loco | $\mathbf{75.95 \pm 0.66}$ | $0.97 \pm 0.01$ | 195 |

Table 24: Low-Impulse Disturbance

| Method | Success Rate ↑ | Goal Tracking ↑ | Switch Count |
|---|---|---|---|
| HWC-Loco-fixed-0.2 | $95.04 \pm 0.31$ | $0.81 \pm 0.01$ | 468 |
| HWC-Loco-fixed-0.4 | $91.81 \pm 0.83$ | $0.92 \pm 0.01$ | 209 |
| HWC-Loco-l | $92.79 \pm 1.75$ | $\mathbf{0.94 \pm 0.01}$ | 47 |
| HWC-Loco | $\mathbf{94.84 \pm 0.54}$ | $0.92 \pm 0.01$ | 198 |

Table 25: High-Impulse Disturbance

| Method | Success Rate ↑ | Goal Tracking ↑ | Switch Count |
|---|---|---|---|
| HWC-Loco-fixed-0.2 | $80.31 \pm 0.54$ | $0.80 \pm 0.01$ | 469 |
| HWC-Loco-fixed-0.4 | $75.84 \pm 0.41$ | $0.90 \pm 0.01$ | 214 |
| HWC-Loco-l | $77.09 \pm 0.79$ | $\mathbf{0.92 \pm 0.01}$ | 61 |
| HWC-Loco | $\mathbf{81.27 \pm 0.80}$ | $0.89 \pm 0.01$ | 215 |

**Robust-optimization Ablation.** We train the recovery policy $\pi_2$ with the ZMP constraint but *without* the extreme-case uncertainty set. As shown in Table 26, using only the ZMP constraint preserves good performance in nominal settings (e.g., low-frequency disturbances), but robustness degrades under challenging conditions such as high-impulse perturbations. This indicates that adversarial training is key for handling real-world uncertainty. Overall, the ZMP constraint promotes baseline safety, while adversarial perturbations are essential for robustness in high-noise or extreme scenarios; the combined effect yields the best performance.

Table 26: Robust-optimization ablation on success rate (%). "w/o" denotes training without the extreme-case uncertainty set.

| Policy | Low-freq. ↑ | Constant ↑ | Low-imp. ↑ | High-imp. ↑ | Low Payload ↑ | High Payload ↑ |
|---|---|---|---|---|---|---|
| HWC-Loco (w/o) | $94.67 \pm 0.31$ | $72.23 \pm 0.43$ | $92.67 \pm 0.75$ | $76.29 \pm 0.34$ | $84.76 \pm 2.66$ | $65.21 \pm 0.83$ |
| **HWC-Loco** | $\mathbf{95.88 \pm 0.37}$ | $\mathbf{75.95 \pm 0.66}$ | $\mathbf{94.84 \pm 0.54}$ | $\mathbf{81.27 \pm 0.80}$ | $\mathbf{87.43 \pm 0.92}$ | $\mathbf{69.86 \pm 1.00}$ |

**Non-robust constrained ablation.** To isolate the effect of the max–min robust optimization term in Eq. 4, we compare HWC-Loco with two non-robust baselines: 1) CRL, a constrained RL variant that optimizes Eq. 2 with the same safety constraint but *without* the adversarial extreme set; and 2) RL-Penalty, a standard RL baseline trained with a large safety penalty under the same domain randomization as CRL. As shown in Table 27, both CRL and RL-Penalty either sacrifice tracking performance or converge to less natural, over-regularized gaits. In contrast, the full robust CRL formulation used by HWC-Loco matches or slightly improves the success rate while achieving better command tracking and more human-like motion. This ablation indicates that the max–min robust optimization in Eq. 4 provides a clear advantage over standard constrained RL and penalty-based reward shaping.

Table 27: Performance comparison with non-robust constrained methods.

| Policy | Success Rate ↑ | Goal-tracking ↑ | Human-like ↓ |
|---|---|---|---|
| CRL | $98.12 \pm 0.32$ | $1.06 \pm 0.08$ | $3.56 \pm 0.14$ |
| RL-Penalty | $96.24 \pm 0.13$ | $1.11 \pm 0.18$ | $3.45 \pm 0.23$ |
| **HWC-Loco** | $\mathbf{98.14 \pm 0.35}$ | $\mathbf{1.12 \pm 0.01}$ | $\mathbf{3.11 \pm 0.03}$ |

**History Length Ablation.** As shown in Table 28, we investigate the impact of observation history length on HWC-Loco's performance. Setting $H = 10$ achieves the best overall trade-off across the three metrics. While increasing $H$ beyond 10 slightly improves the success rate, it leads to a

noticeable decline in goal-tracking accuracy and human-likeness. This suggests that excessively long history may hinder the effective use of privileged information, resulting in more conservative and less adaptive behaviors.

Table 28: Performance comparison with different history lengths ($H$)

| History Length | Success Rate ↑ | Goal-tracking ↑ | Human-like ↓ |
|---|---|---|---|
| 1 | $95.13 \pm 0.51$ | $1.07 \pm 0.00$ | $3.26 \pm 0.06$ |
| 10 | $96.73 \pm 0.40$ | $\mathbf{1.10 \pm 0.00}$ | $\mathbf{3.18 \pm 0.01}$ |
| 20 | $96.53 \pm 0.40$ | $1.08 \pm 0.00$ | $3.18 \pm 0.02$ |
| 40 | $\mathbf{97.38 \pm 0.33}$ | $1.02 \pm 0.00$ | $3.21 \pm 0.02$ |

## B.6 REAL WORLD EXPERIMENTS

**Effectiveness Tests in the Real World.** Figure 10 demonstrates the robot's ability to traverse a multi-terrain course consisting of 15 cm steps, 20° slopes, and wooden boards at a speed of 0.3 m/s. Figure 11 shows the ability to stability in real-world outdoor locomotion, including flat ground, grassy surfaces, and slopes.

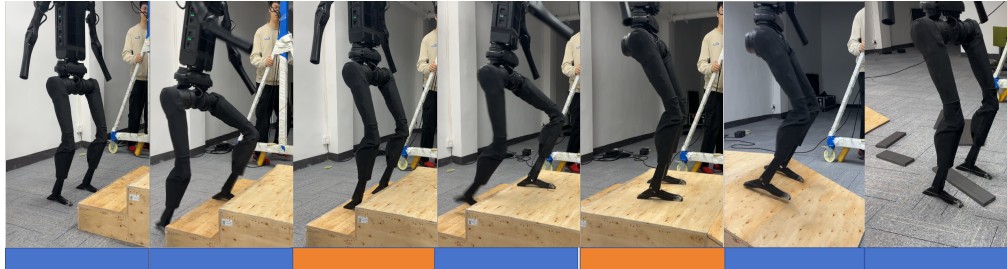

Figure 10: Climb Stairs Test. The blue segments indicate the activation of the goal-tracking policy, while the orange segments correspond to the safety recovery policy.

**Robustness Tests in the Real World.** As shown in Figure 12, we apply various disturbances to Unitree H1 lasting for one minute. Four representative disturbance segments were selected for analysis. The time is measured in control cycles, with each cycle lasting 0.01 seconds. Furthermore, we conduct an outdoor disturbance test by applying pushes, pulls, and kicks to the robot, as shown in Figure 13. The robot waves its arms and adjusts its gaits to maintain balance. Notably, the high control frequency (100 Hz) ensures smooth transitions between policies, and isolated activations of the recovery policy have minimal impact on overall stability.

**Failure Mode Analysis.** Although HWC-Loco is robust across diverse scenarios, it is not failure-free. In real-world tests, typical failures arise when operating conditions lie far outside the training distribution:

- **Extreme surface conditions.** On very low-friction surfaces (e.g., ice-like friction far below training values), the feet cannot generate sufficient ground reaction forces, and the robot may slip and fall even when $\pi_0$ frequently invokes $\pi_2$.

- **Aggressive, out-of-range commands.** When commanded velocities or abrupt turns significantly exceed the training range, $\pi_1$ can drive the robot into configurations from which $\pi_2$ can no longer recover.

- **Unmodeled hardware issues.** Severe communication delays, sensor drop-outs, or actuator faults are not fully captured by our uncertainty set; in such cases, violated feedback assumptions can cause failures despite switching to $\pi_2$.

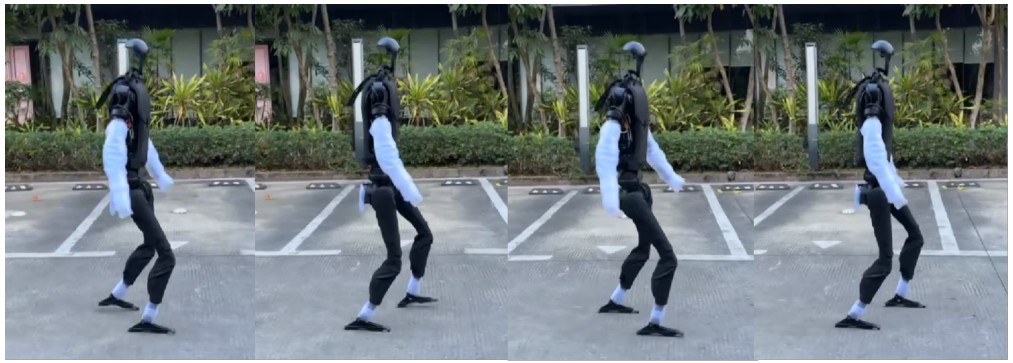

(a) Walking on flat terrain.

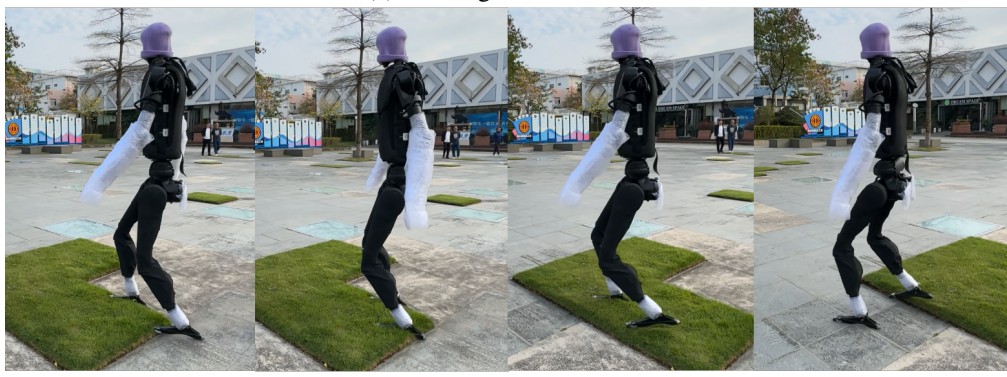

(b) Traversing grassy surfaces.

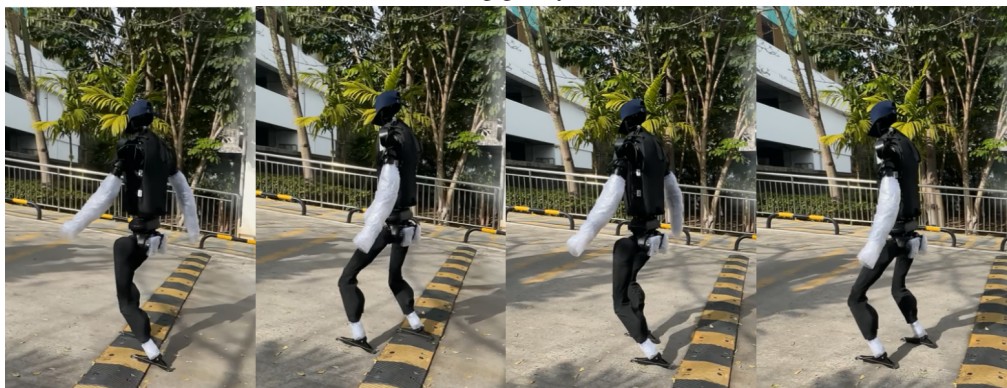

(c) Climbing a slope.

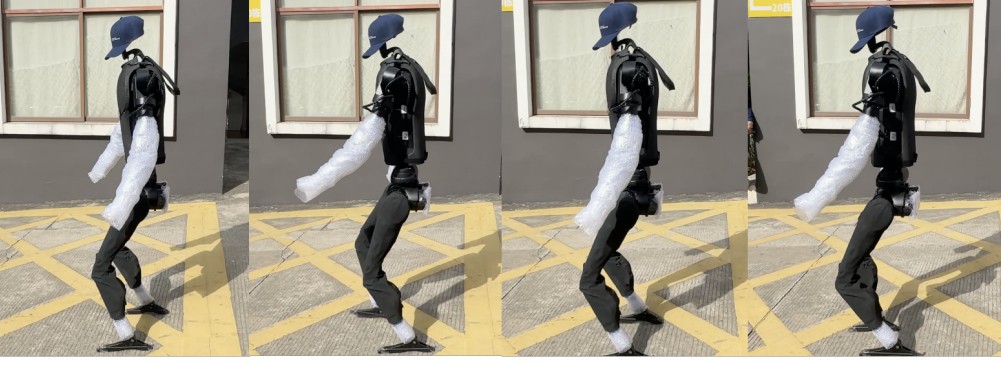

(d) Descending a slope.

Figure 11: Outdoor Deployment across Diverse Terrains: (a) flat ground, (b) grassy surface, (c) upward slope, and (d) downward slope.

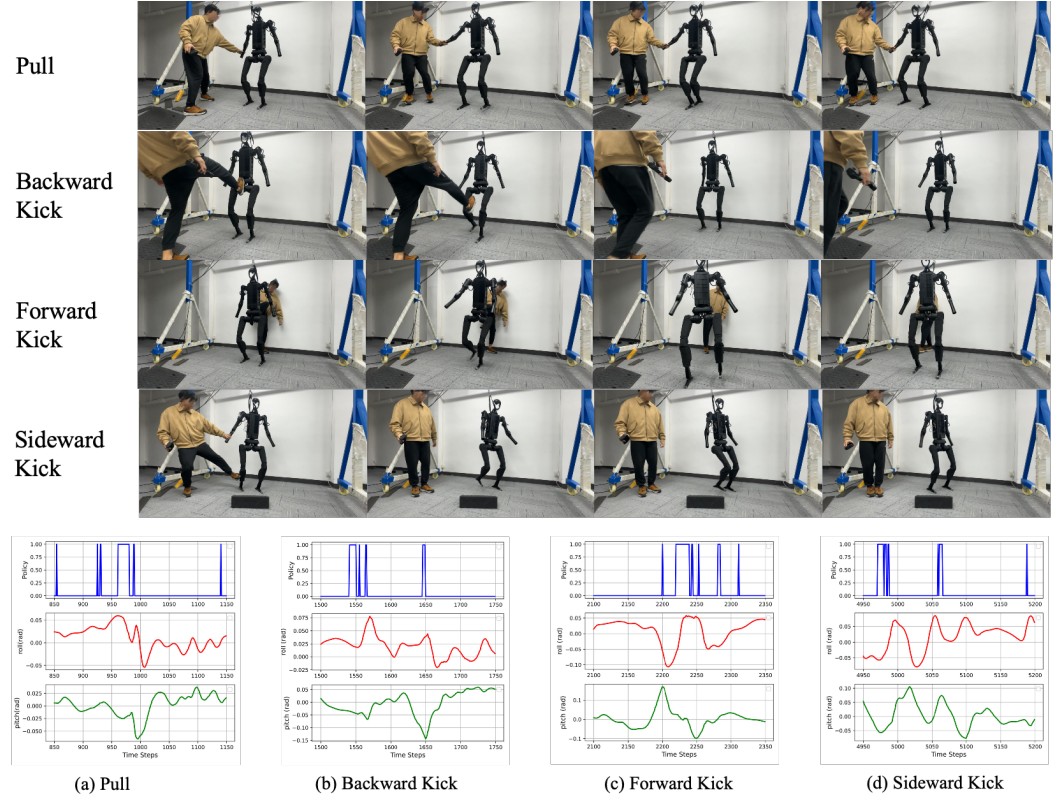

Figure 12: Disturbances in Realistic Deployment: Policy Switching value of 0 corresponds to the goal-tracking policy, while a value of 1 denotes the safety recovery policy.

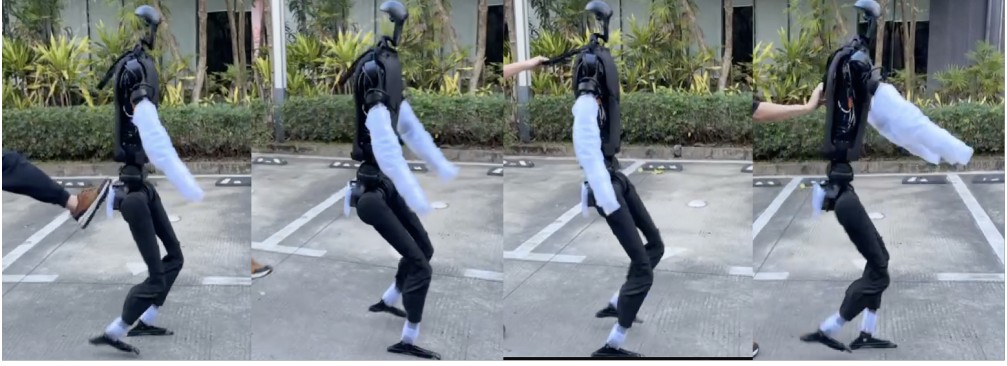

Figure 13: Robustness in Outdoor Settings: The robot responds to external disturbances in an outdoor environment by waving its arms and adjusting its gaits to maintain balance.

### B.7 SCALABILITY EXPERIMENTS

**Case Study on Motion Tracking.** We visualize the robot's performance on tracking a punching motion in Figure 14. During the execution, the robot experiences an impulse impact. In response, it dynamically activates the recovery policy, switching to more conservative and safer actions. Specifically, it lowers its body, adjusts its gait, and stabilizes itself to satisfy ZMP constraints, rather than strictly following the original target motion. As a result, it generates a backtracking motion to maintain balance and ensure robustness.

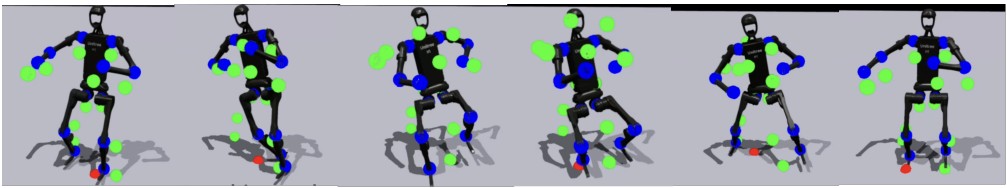

Figure 14: Robustness in Motion Tracking: The green dots represent the target keypoints, the blue dots indicate the keypoints corresponding to the DoFs, and the red dot denotes the ZMP.

Table 29: Comparison of Humanoid Configurations

| Parameter | H1 | G1 |
|---|---|---|
| Height | 178 cm | 127 cm |
| Mass | 47 kg | 35 kg |
| DoFs | 19 | 23 |

**General and Flexible Locomotion.** G1 outperforms H1 across all three metrics (success rate, goal-tracking accuracy, and human-likeness) primarily due to the following factors:

- **Size and Mass:** As shown in Table 29, G1's smaller size and lighter mass contribute to lower inertial loads and reduced torque demands. This facilitates greater stability, especially during dynamic motions, and enables faster corrective actions in response to disturbances.

- **Degrees of Freedom (DoFs):** With 23 DoFs compared to H1's 19, G1 offers enhanced joint-level flexibility. The additional DoFs, particularly in the upper body and feet, allow for better whole-body coordination. This increased redundancy leads to improved disturbance rejection and more human-like behavior during complex locomotion tasks.

Notably, as illustrated in Figure 15, G1 is capable of achieving straight-knee walking, which remains a significant challenge for H1 due to its larger size and reduced agility. The additional DoFs in G1 enable finer control of limb trajectories, which is critical for maintaining balance during such biomechanically demanding gaits.

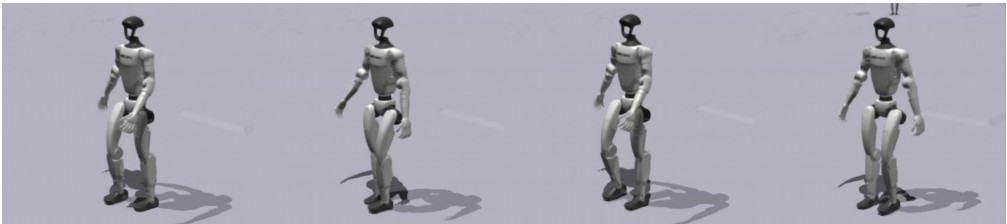

Figure 15: G1 walk with straight knees

**Robust Balance under Disturbances.** To qualitatively evaluate balance and recovery in challenging motions, we deploy HWC-Loco on the Unitree G1 and command a demanding single-leg stance under external pushes, using motion data from HuB Zhang et al. (2025a). As shown in Fig. 16, the robot is able to maintain one-leg balance while compensating for the induced disturbances through

coordinated whole-body responses. By leveraging its safety-aware structure to preserve a stable ZMP region and avoid catastrophic falls, HWC-Loco demonstrates superior robustness in safety-critical regimes.

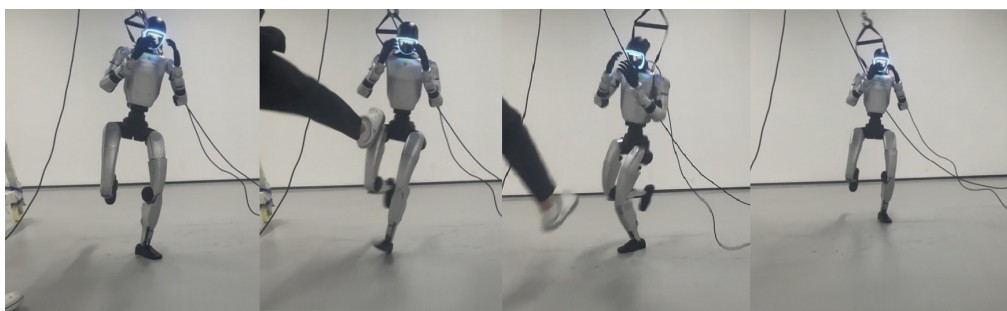

Figure 16: Single-leg balancing under external disturbances.

