# OpenReview forum: "HWC-Loco: A Hierarchical Whole-Body Control Approach to Robust Humanoid Locomotion"
_ICLR.cc/2026/Conference — ICLR 2026 Poster_

### Official Review · Reviewer_tg3Q · 2025-10-25

**Soundness:** 2
**Presentation:** 3
**Contribution:** 3
**Rating:** 4
**Confidence:** 4

**Summary:**

This paper presents HWC-Loco, a hierarchical whole-body control framework for robust humanoid locomotion. It combines a goal-tracking policy for task performance and human-like motion with a safety policy for handling safety-critical scenarios with a high-level planner that can dynamically switches between them to keep safe and precise. By formulating policy learning as a robust optimization problem under mismatched environmental dynamics, HWC-Loco ensures safety while maintaining efficiency, demonstrating superior performance across diverse terrains, robots, and tasks in both simulation and real-world deployments.

**Strengths:**

1. HWC-Loco quantitatively improves task success rates and demonstrates reasonable high-level policy switching to maintain safety.
2: The high-level policy and hierarchical architecture are well-designed, effectively balancing goal-tracking and safety recovery.
3. The paper presents extensive experiments, and the overall structure and clarity of the manuscript are strong.

**Weaknesses:**

1. The paper lacks comparison to commonly used baselines such as larger domain randomization or larger domain randomization combined with history-aware policies. While the authors argue that “excessive regularization can greatly affect the efficiency of control policy,” they do not provide evidence that HWC-Loco achieves a better trade-off between efficiency and safety relative to these methods. (If these issues are addressed with stronger evidence, I’d be happy to raise my score.)
2. Performance/Evaluation Concerns: 1) The terrain experiments do not clearly illustrate when or how the high-level policy switching occurs, making it difficult to understand the specific contribution of the safety policy. 2) The terrains and challenge cases used are relatively simple. The paper should include scenarios that highlight the necessity of the hierarchical design—i.e., cases where success cannot be achieved without it or with standard domain randomization alone. Providing qualitative comparisons would help distinguish HWC-Loco from these baselines.
3. The reliance on ZMP constraints is not that generalizable or robust. ZMP-based control depends heavily on accurate robot dynamics modeling and sensing, which can undermine the reliability (though I agree that integrating it within hierarchical, rather than enforcing hard constraints, is a more flexible design). Moreover, for tasks involving rapid or aerial motions (e.g., jumping) or requiring more agile locomotion, enforcing ZMP-based safety objectives may conflict with task goals, thereby restricting the effectiveness of the safe policy. Finally, its applicability to loco-manipulation under external forces remains limited

**Questions:**

1. Why does the baseline Goal-tracking consistently outperform AHL/DreamWaQ?
2. Some videos on the project website (e.g., Soft and Slippy Terrain) were not visible at my review time. Is this an accessibility issue on my side, or were those videos not yet uploaded?

---

> ### Author Response · Authors · 2025-11-22
>
> Dear Reviewer tg3Q, we sincerely appreciate the time and effort you’ve invested in evaluating our work. We have prepared comprehensive responses and clarifications to address each point you raised. We hope these responses can resolve your concerns.
>
> ---
> >1. W1: Comparison to large domain randomization with history-aware policies.
> ---
> Thanks for raising this important point. Our existing baselines already include domain-randomized, history-aware policies such as AHL. To more clearly disentangle the effect of **larger domain randomization** from the benefit of our hierarchical design, we additionally introduce a **strong non-hierarchical baseline** (evaluated at randomization scales 1, 2, and 4) that uses a single history-aware policy trained with large-scale domain randomization and regularization rewards, denoted **Large-DR-Hist**. This baseline shares the same randomized dynamics and disturbance distributions as the safety-recovery policy $ \pi_2 $ (e.g., masses, friction, payloads, delays, external pushes). However, as we substantially increase the randomization ranges, optimization becomes noticeably harder and the resulting policy tends to adopt overly conservative gaits.
>
> We compare HWC-Loco and Large-DR-Hist across general locomotion tasks with diverse terrains and randomized dynamics. The results are as follows:
>
> | **Policy** | **Success Rate** ↑ | **Goal-tracking** ↑ | **Human-like** ↓ |
> |-|-|-|-|
> | Large-DR-Hist (1.0) | 96.24 ± 0.13 | 1.11 ± 0.21 | 3.57 ± 0.23 |
> | Large-DR-Hist (2.0) | 96.56 ± 0.26 | 1.09 ± 0.34 | 3.61 ± 0.11 |
> | Large-DR-Hist (4.0) | **98.18 ± 0.45** | 1.02 ± 0.32 | 3.68 ± 0.31 |
> | **HWC-Loco** | 98.14 ± 0.35 | **1.12 ± 0.01** | **3.11 ± 0.03** |
>
> As shown in the table, Large-DR-Hist exhibits the typical behavior of heavily domain-randomized single policies. As the **randomization range increases**, success rate improves slightly, but the policy becomes increasingly **over-regularized**, leading to overly cautious gaits with reduced speed, hesitant stepping, and less human-like motion. This highlights the difficulty of optimizing a single controller over a broad and challenging dynamics distribution.
>
> In contrast, HWC-Loco achieves a **more favorable efficiency–safety trade-off**, attaining comparable (or higher) success rates while preserving sharper command tracking and more natural, human-like motion. HWC-Loco can execute tasks efficiently under nominal conditions, while selectively prioritizing stability via the safety-recovery policy when needed. These results underscore the advantage of our hierarchical robust CRL framework over a single history-aware large-DR policy in simultaneously delivering efficiency and robustness.
>
> ---
> >2. W2: Clarifying when and how the high-level policy switches, and the necessity of the hierarchy.
> ---
> Thanks for this valuable feedback. Our current switching visualizations are mainly in the appendix (e.g., **Fig. 10, Fig. 12**) and demo videos, where time-series plots show commanded velocity, VAE-estimated stability features (e.g., ZMP deviation), and the discrete decisions of $ \pi_0 $. They indicate that $ \pi_0 $ switches to $ \pi_2 $ when the estimated stability margin deteriorates (e.g., after a strong push or slip), and returns to $ \pi_1 $ once stability is restored; we will make this more explicit in the main text and figure captions.
>
> When the robot aggressively tracks commands (fast walking/turning) under large disturbances, single-policy baselines—even with strong domain randomization—tend to keep tracking and eventually fall. In contrast, HWC-Loco uses $ \pi_0 $ to temporarily hand control to $ \pi_2 $ for recovery, and then switches back to $ \pi_1 $ once the robot is stable again. This mechanism leads to substantially higher success rates in robustness tests (Table 2), and is further illustrated by case studies in Figures 12–14 and the videos (e.g., “Hard Kick” and “Malicious Commands”). We also include a demo ("Standard RL") where a standard domain-randomized and regularized RL policy is subjected to a hard impact while walking or running: it fails to recognize the hazardous situation and cannot recover, in clear contrast to our hierarchical controller. Together, these results demonstrate that the hierarchy is practically necessary to handle strong disturbances without sacrificing nominal performance.

---

> > ### Author Response · Authors · 2025-11-22
> >
> > ---
> > >3. W3: Limitations of ZMP-based constraints and generality.
> > ---
> > Thanks for raising this point. Our claims are explicitly scoped to **walking and moderately fast running with continuous ground contact**, where ZMP remains a meaningful proxy for stability; we do not claim to directly handle highly dynamic aerial behaviors or complex loco-manipulation under large external forces. To mitigate known ZMP limitations, we (i) treat it as a **soft, learned feasibility objective** for the recovery policy $ \pi_2 $ rather than a hard analytical constraint, (ii) randomize dynamics and **sensor noise/latency** so that $ \pi_2 $ learns to cope with imperfect ZMP estimates, and (iii) confine ZMP-based objectives mainly to $ \pi_2 $, while $ \pi_1 $ focuses on task performance and human-like motion, with $ \pi_0 $ invoking $ \pi_2 $ only when stability degrades.
> >
> > ---
> > >4. Q1: Why does the goal-tracking baseline outperform AHL/DreamWaQ?
> > ---
> > Thanks for this question. In our setup, the goal-tracking baseline replaces strong gait/pose regularization with a Wasserstein imitation objective on our human-motion dataset, which encourages more flexible, task-specific locomotion and yields better nominal tracking and motion naturalness than our adaptations of AHL and DreamWaQ (which were originally tuned for different robots and objectives). In contrast, AHL and DreamWaQ rely on relatively strong regularization and conservative design choices to improve expressiveness and safety, which can reduce tracking sharpness and robustness in our setting.
> >
> > Importantly, our goal-tracking baseline does not dominate these methods on all robustness tests (see Table 2), highlighting that single-policy controllers naturally occupy different points on the efficiency–robustness trade-off. HWC-Loco instead leverages a hierarchy to improve robustness without sacrificing nominal performance.
> >
> > ---
> > >5. Q2: Project videos accessibility.
> > ---
> > Thanks for pointing this out. We have double-checked that all project videos are correctly uploaded and accessible.

---

> > ### Comment · Reviewer_tg3Q · 2025-11-26
> > **Response to the authors**
> >
> > Thanks for the additional experiments and valuable feedback; however, although HWC-Loco does report improvements over domain-randomized baselines, many of the gains are numerically marginal and, in several cases, fall within variances. This makes it difficult to conclude that the observed differences are practically meaningful.
> >
> > For robotics and safety-oriented claims, I would strongly prefer to see **real-world evidence**, either through qualitative demonstrations (e.g., videos showing significantly **better** behaviors under disturbances compared to baselines) or quantitative metrics. Without such real-world validation or sufficiently robust simulation support, it remains unclear whether the reported improvements would persist or be overshadowed by hardware noise or unmodeled disturbances. If real-world evidence is unavailable, the work should instead provide **sufficiently strong and convincing** (significant improvement that should remain meaningful despite real-world noise) simulation results demonstrating a significantly higher performance upper bound than the baselines.
> >
> > Consequently, it is difficult to be fully convinced that switching locomotion control from a domain-randomized (DR) policy to HWC-Loco is necessary, especially given that DR already provides a more **general**, broadly applicable, and simpler alternative, whereas HWC-Loco remains limited to “walking and moderately fast running with continuous ground contact.”
> >
> > So I will keep my score.

---

> > > ### Author Response · Authors · 2025-12-03
> > >
> > > ---
> > > >1. Performance in simulation results.
> > > ---
> > > Thanks for this insightful concern. We apologize for not stating this more clearly. In the previous additional experiments, all policies were evaluated in a general locomotion setting where terrains and disturbances remained within the training distribution (consistent with Table 3). In this regime, our goal is not to demonstrate dramatic safety gains, but to show that HWC-Loco achieves a **better robustness–efficiency trade-off** than DR baselines: it maintains competitive nominal performance while improving robustness.
> > >
> > > However, our **safety-recovery claim** is primarily targeted at **extreme scenarios**, where disturbances and noise are large and potentially dangerous, rather than at nominal conditions that standard domain randomization already handles reasonably well. To address this, we have added new comparisons between HWC-Loco and domain-randomized baselines under such extreme perturbations. In these extreme settings, HWC-Loco exhibits **clearly superior performance**, with substantially fewer catastrophic failures and more successful recoveries. We believe these results more directly reflect the benefit of the proposed safety-critical hierarchy on top of domain randomization.
> > >
> > > | **Policy**| **External Force/Torque Disturbances** || **Impulse Disturbances on CoM** || **Payload on Upper Body** ||
> > > |-|-|-|-|-|-|-|
> > > || **Low-freq. ↑**| **Constant ↑** | **Low-imp. ↑**| **High-imp. ↑** | **Low Payload ↑** | **High Payload ↑** |
> > > | Large-DR-Hist (1.0)| 87.15 ± 0.23| 61.72 ± 0.36| 85.45 ± 0.87| 62.38 ± 1.03| 79.53 ± 0.76| 60.23 ± 0.61|
> > > | Large-DR-Hist (2.0)| 88.45 ± 0.56| 65.32 ± 0.28| 88.47 ± 1.75| 68.36 ± 0.79| 82.32 ± 0.87| 65.21 ± 0.57|
> > > | Large-DR-Hist (4.0)| 90.35 ± 0.43| 70.53 ± 0.42| 90.67 ± 1.31| 71.36 ± 0.89| 85.77 ± 0.93| **70.36 ± 0.69**|
> > > | **HWC-Loco**| **95.88 ± 0.37**| **75.95 ± 0.66** | **94.84 ± 0.54**| **81.27 ± 0.80** | **87.43 ± 0.92** | 69.86 ± 1.00|
> > >
> > > ---
> > > > 2. Real-world relevance and robustness to hardware noise.
> > > ---
> > > Thank you for raising this important point. In practice, however, systematically applying extreme disturbances (e.g., large pushes, severe slips) on full-scale humanoids is both risky and costly, and external perturbations such as pushes, pulls, and terrain irregularities are difficult to apply in a precise, repeatable, and comparable way across controllers. This makes it hard to construct fair and fully quantitative comparison tables on real hardware.
> > >
> > > For this reason, we choose to **quantify extreme robustness primarily in simulation** (Table 2), where we can carefully sweep disturbance magnitudes and configurations and report comprehensive statistics that approximate the extreme scenarios likely to occur in the real world. For real-world validation, we have conducted extensive robustness tests on a Unitree H1 humanoid (random command tracking, diverse terrains, external push/pull). In addition, we deploy HWC-Loco on a **Unitree G1 humanoid** and release further G1 demos (e.g., Single Leg Balance) on our anonymous project website ([HWC-Loco Demo](https://anonymous.4open.science/w/HWC_Loco_demo/)). These real-world results qualitatively confirm that the robustness observed in simulation carries over to hardware under realistic noise and disturbances.
> > >
> > > ---
> > > >3. Why use HWC-Loco instead of a purely DR policy?
> > > ---
> > > Thanks for raising this concern. We would like to clarify that HWC-Loco is **not intended as an alternative to domain randomization (DR)**. In fact, all of our policies are trained with some of the standard DR techniques (e.g., sensor noise, disturbances). Our goal is to **decouple safety-critical scenarios from nominal, generally safe scenarios** and let different components specialize accordingly.
> > >
> > > Concretely, DR policies for locomotion are typically optimized with complex reward functions whose core terms are **task-centric** (e.g., motion/command/position tracking). As a result, they naturally prioritize tracking performance, and safety-recovery often appears only as a secondary effect emerging from these reward terms.
> > >
> > > We therefore view HWC-Loco as a **structurally safety-aware** extension of DR, rather than an alternative:
> > >
> > > - Our DR baselines are already strong and well-tuned, but they remain largely **agnostic to structural stability constraints**, such as ZMP-based margins. This can lead them to exploit narrow regions of the dynamics where tracking performance is high, but safety margins are small, resulting in brittle behavior under unexpected perturbations.
> > >
> > > - HWC-Loco explicitly couples a **high-level policy** with a **safety-oriented controller** that enforces stability and recovery behaviors, while still operating under comparable domain randomization. This hierarchical design is precisely what yields **fewer catastrophic failures and more successful recoveries** in disturbed scenarios, without sacrificing competitive performance in nominal conditions.

---

### Official Review · Reviewer_qF8c · 2025-10-29

**Soundness:** 3
**Presentation:** 3
**Contribution:** 2
**Rating:** 4
**Confidence:** 3

**Summary:**

The paper proposes HWC-Loco, a hierarchical approach for humanoid locomotion balancing safety and agility. The policy is switching between two policy classes: a goal-tracking policy which imitate human motion and a recovery policy trained with robust objective. Then, a high-level planner is learned to switch between two policies optimizing for overall tracking performance. The results is verified in sim and real world.

**Strengths:**

- the hierarchical approach that target for safety and performance trade-off is natural and well-motivated.

- the performance of controller is evaluated comprehensively with diverse metrics to show is robustness and naturalness.

**Weaknesses:**

- comparison mainly ephasize dreamwaq and hal, where recent strong baselines for humanoid locomotion without hierarchical design like [1] as well as the same style switch controller like [2] is missing.

- it would be beneficial if the author could further discuss the sim2real gap identified in the real world deployment and explains with quantitative result on how those sensing and actuation gap hurts the standard policy design and why the hierarchical design can improve on it.


[1] AdaMimic: Towards Adaptable Humanoid Control via Adaptive Motion Tracking, Huang et al

[2] Agile But Safe: Learning Collision-Free High-Speed Legged Locomotion, He et al.

**Questions:**

- how sensitive the tracking/robustness trade off to the imitation multiplier lambda? would the increase of human-like behavior would lead to less recovery agility?

- is it possible to distill two policies into a single one to enable smooth transition by teacher-student training?

---

> ### Author Response · Authors · 2025-11-22
>
> Dear Reviewer qF8c, we truly appreciate the time and effort you’ve invested in evaluating our work. We have prepared comprehensive responses and clarifications to address each point you raised. We hope these responses can resolve your concerns.
>
> ---
> >1. W1: Stronger baselines comparisions.
> ---
> Thanks for highlighting these related works. **AdaMimic** focuses on high-fidelity adaptive imitation and **Agile But Safe (ABS)** on safety-aware high-speed navigation. Their ideas are conceptually close to ours, but their systems are designed for different morphologies, tasks, and sensing, and are hard to port to our whole-body, multi-terrain humanoid setup without substantial redesign; public code and hyperparameters also do not match our evaluation suite. To avoid unfair or unstable reimplementations, we instead adopt **two recent, strong, reproducible baselines** that can be directly adapted to our environments:
>
> 1. Narrow [1]: ZMP-based rewards and angular-momentum regularization for stable narrow-footprint walking.
> 2. Perceptive [2]: terrain-aware locomotion with a learned terrain encoder and world model.
>
> Adapted to our setting and evaluated under the same disturbance suite, we obtain:
>
> | **Policy**| **External Force/Torque Disturbances** || **Impulse Disturbances on CoM** || **Payload on Upper Body** ||
> |-|-|-|-|-|-|-|
> || **Low-freq. ↑**| **Constant ↑** | **Low-imp. ↑**| **High-imp. ↑** | **Low Payload ↑** | **High Payload ↑** |
> | Narrow| 88.04 ± 0.36| 67.67 ± 0.34| 89.18 ± 0.92| 75.64 ± 0.83| 81.53 ± 0.76| 62.73 ± 0.43|
> | Perceptive| 95.45 ± 0.56| 70.32 ± 0.28| 90.47 ± 1.75| 72.36 ± 0.79| 80.52 ± 0.87| 59.21 ± 0.59|
> | **HWC-Loco**| **95.88 ± 0.37**| **75.95 ± 0.66** | **94.84 ± 0.54**| **81.27 ± 0.80** | **87.43 ± 0.92** | **69.86 ± 1.00**|
>
> HWC-Loco achieves the highest success rates across all disturbance types, indicating that hierarchical robust constrained training provide clear benefits over strong baselines. In the revised manuscript, we have briefly related our design to AdaMimic and ABS in Related Work and discuss the results with recent strong baselines.
>
> **References:**
> [1] Xie W, Bai C, Shi J, et al. *“Humanoid Whole-Body Locomotion on Narrow Terrain via Dynamic Balance and Reinforcement Learning.”* arXiv, 2025.
> [2] Sun W, Cao B, Chen L, et al. *“Learning Perceptive Humanoid Locomotion over Challenging Terrain.”* arXiv, 2025.
>
> ---
> >2. W2: Quantifying Sim2Real gap and the benefit of the hierarchy.
> ---
> Thanks for raising this point. A concrete Sim2Real issue appears in **stair climbing**. In simulation, both the goal-tracking baseline and HWC-Loco can climb stairs reliably. On the real robot, however, sensing and contact-model mismatch sometimes cause the swing foot to land near the stair edge or partially miss the step. A pure goal-tracking policy $ \pi_1 $ keeps leaning forward to follow the command, which often turns this small error into a fall. With the hierarchy, $ \pi_0 $ detects the degraded stability margin in such events and temporarily switches to the recovery policy $ \pi_2 $ to re-stabilize the robot (e.g., by adjusting CoM and stepping), before handing control back to $ \pi_1 $. This mechanism quantitatively reduces real-world failures on stairs compared to the standard policy.

---

> > ### Author Response · Authors · 2025-11-22
> >
> > ---
> > >3. Q1: Sensitivity to the imitation weight λ.
> > ---
> > Thanks for raising this point. The imitation multiplier $\lambda$ sets the strength of the Wasserstein imitation objective for $\pi_1$. We use a nominal value and perform a small sweep around it, experimenting with scales 0.5, 1.0, and 2.0, where 1.0 is the original setting.
> >
> > As shown in the table, smaller $\lambda$ improves raw command-tracking but results in less human-like, stiffer motion. Larger $\lambda$ generates smoother, more human-like gaits, albeit with a modest increase in tracking error. Once combined with the **same** $\pi_2$ and $\pi_0$, robustness under disturbances changes only marginally across this range, since recovery capability is mainly governed by $\pi_2$ and the switching behavior of $\pi_0$. Therefore, HWC-Loco is not overly sensitive to this parameter.
> >
> >
> > | **Policy**| **Success Rate** ↑| **Goal-tracking** ↑| **Human-like** ↓|
> > |-|-|-|-|
> > | HWC-Loco (0.5)| **98.21 ± 0.14**| 1.12 ± 0.01| 3.23 ± 0.03|
> > | HWC-Loco (1.0)| 98.14 ± 0.35| **1.12 ± 0.01**| 3.11 ± 0.03|
> > | HWC-Loco (2.0)| 98.08 ± 0.26| 1.12 ± 0.02**| **3.08 ± 0.03**|
> >
> > ---
> > >4. Q2: Distilling the two policies into a single network.
> > ---
> > Thanks for this insightful suggestion. In principle, the hierarchy could act as a *teacher* and be distilled into a single *student* policy. In our setting, however, $ \pi_1 $ and $ \pi_2 $ are trained for **different objectives** under **different dynamics distributions**, and a single student that tries to imitate both tends to become either as conservative as $ \pi_2 $ or as fragile as $ \pi_1 $, depending on the distillation weights.
> >
> > A more promising direction is a **conditional or MoE-style student**, whose internal experts specialize in nominal tracking vs. recovery, with a learned gating module replacing the external switch. Systematically exploring such distillation schemes is beyond the scope of this work, but we will briefly mention this as an interesting direction for future research.

---

> > > ### Comment · Reviewer_qF8c · 2025-11-23
> > > **Thanks for the comprehensive response**
> > >
> > > Thank you for the comprehensive response and additional experiments.
> > >
> > > I appreciate the new "narrow" and "perceptive" experiments. The comparisons address my concerns on method's performance. I am also convinced about the imitation multiplier design.
> > >
> > > With that said, I will raise my score to weak accept.

---

### Official Review · Reviewer_qydQ · 2025-11-01

**Soundness:** 4
**Presentation:** 4
**Contribution:** 4
**Rating:** 8
**Confidence:** 4

**Summary:**

The authors present a learning based hierarchical control framework for humanoid robots that ensures task completion while maintaining safety requirements. A high level planning policy, trained using double DQN, selects between a goal tracking policy and a safety recovery policy based on robots state and historical observations. The paper presents impressive results in simulation as well as real world experiments, across various tasks and terrains. The extensive evaluation demonstrates superior performance over comparable baselines.

**Strengths:**

The paper is well written, structured and easy to understand. The paper tackles a challenging problem and the proposed approach is appropriately motivated and positioned well among related work. Real world robust humanoid locomotion is a challenging task and the results presented in this paper is quite impressive.

The paper provides sufficient experimental details and extensive validation. Tests across a wide range of conditions such as terrains, task commands and disturbances provide insightful details about the policies performance. Comparisons against relevant baselines and ablation studies shed light on the effectiveness of the proposed approach well.

**Weaknesses:**

One of the weaknesses I can think of is the complex framework - this involves training two lower level policies separately, then a high level planner , GAN discriminator, and to enable real world deployment a VAE encoder that estimates privileged information that the policy has during training in simulation. These design choices might be hard to reproduce or deploy.

Since the networks are not trained jointly, there could be switching behavior between the two low level policies due to errors in the system. For example, if the VAE estimator has noisy estimates of privileged information.

**Questions:**

1) What are some of the failure cases and what caused them?
2) What sort of domain randomization was applied in simulation?

---

> ### Author Response · Authors · 2025-11-22
>
> Dear Reviewer qydQ, we truly appreciate the time and effort you’ve invested in evaluating our work. We have prepared comprehensive clarifications in the hope of addressing your concerns.
>
> ---
> >1. W1: On the complexity and reproducibility of the framework.
> ---
> Thanks for raising this concern. We agree that HWC-Loco has multiple components, but each is built from **standard modules**: PPO for $ \pi_1 $/$ \pi_2 $, a discriminator for the Wasserstein imitation loss, Double-DQN for the high-level planner π₀, and a lightweight VAE for privileged-information estimation.
>
> Training is **staged and decoupled**, which we have found to simplify implementation and debugging:
> 1. Train the goal-tracking policy $ \pi_1 $ with PPO + Wasserstein imitation.
> 2. Train the safety-recovery policy $ \pi_2 $ under the robust constrained RL objective (Eq. 4) using the same PPO backbone.
> 3. Train the high-level planner π₀ (Double-DQN) on top of the low-level policies.
>
> We will also release pretrained models upon acceptance to further lower the barrier to reproducing and deploying the framework.
>
> ---
> >2. W2: On potential chattering due to VAE estimation noise.
> ---
> Thanks for this insightful comment. We mitigate chattering from estimation noise in two ways:
>
> 1. **Switching penalty in π₀’s reward.** π₀’s reward includes an explicit penalty on switches between $ \pi_1 $ and $ \pi_2 $, discouraging rapid toggling.
> 2. **Training π₀ on noisy estimates.** π₀ is **always** trained on VAE-estimated privileged features (not ground truth), so it learns under the same noise that appears at test time.
>
> We further add two quantitative analyses.
>
> **VAE estimation accuracy across disturbances.**
>
> | Scenario      | Velocity ↓        | ZMP Features ↓        |
> |---------------|-------------------|------------------------|
> | Normal        | 0.0250 ± 0.0008   | **0.0491 ± 0.0005**   |
> | Low-Frequency | **0.0242 ± 0.0009** | 0.0493 ± 0.0008     |
> | Constant      | 0.0294 ± 0.0009   | 0.0507 ± 0.0009       |
> | Low-Impulse   | 0.0408 ± 0.0013   | 0.0518 ± 0.0016       |
> | High-Impulse  | 0.0719 ± 0.0014   | 0.0531 ± 0.0014       |
> | Low Payload   | 0.0497 ± 0.0012   | 0.0577 ± 0.0012       |
> | High Payload  | 0.0510 ± 0.0013   | 0.0751 ± 0.0018       |
>
> **Robustness to injected estimation noise.**
>
> | Noise Level σ | Success Rate (%) ↑ | Flip Count ↓ |
> |---------------|--------------------|--------------|
> | 0.0           | **78.65 ± 0.53**   | 80           |
> | 0.1           | 77.77 ± 0.62       | 81           |
> | 0.2           | 77.36 ± 0.61       | 82           |
> | 0.5           | 77.33 ± 0.64       | 84           |
> | 1.0           | 76.99 ± 0.83       | 96           |
> | 2.0           | 21.81 ± 3.41       | **126**      |
>
> For all **realistic** noise levels (σ ≤ 1.0), success rates stay high (>76%) and Flip Count increases only mildly; degradation appears only under extreme noise (σ = 2.0), well beyond the observed VAE error. This indicates that π₀ is robust to realistic estimation noise and does not exhibit significant chattering. These results are included in the revised manuscript.
>
> ---
> >3. Q1: What are some of the failure cases and what caused them?
> ---
> Thanks for raising this insightful comment. Although HWC-Loco is substantially more robust than all baselines we tested, it is not failure-free. Typical failures occur when conditions lie well outside the training distribution:
>
> 1. **Extreme surface conditions.**
>    On very low-friction surfaces (e.g., ice-like friction far below training values), feet cannot generate sufficient ground reaction forces, and the robot may slip and fall even when $ \pi_0 $ frequently invokes $ \pi_2 $.
>
> 2. **Aggressive, out-of-range commands.**
>    When commanded velocities or abrupt turns significantly exceed the training range, $ \pi_1 $ can push the robot into configurations from which $ \pi_2 $ cannot recover.
>
> 3. **Unmodeled hardware issues.**
>    Severe communication delays, sensor drop-outs, or actuator faults are not fully captured by our uncertainty set; in such cases, broken feedback assumptions can cause failures despite switching to $ \pi_2 $.
>
> We describe these failure modes in a short subsection of the revised version to clarify the current limitations of HWC-Loco.
>
> ---
> >4. Q2: What sort of domain randomization was applied in simulation?
> ---
> Thanks for raising this point. We apologize for not detailing this more clearly. During training, we apply domain randomization to model Sim2Real mismatch and disturbances. The full settings (ranges for masses, friction, payloads, delays, pushes/impulses, noise, etc.) are provided in Appendix A.6 (Table 12).

---

> > ### Comment · Reviewer_qydQ · 2025-11-23
> >
> > Thank you for the detailed reply. The additional information does bring further clarity in the paper. I will keep my score.

---

### Official Review · Reviewer_L28V · 2025-11-01

**Soundness:** 3
**Presentation:** 3
**Contribution:** 3
**Rating:** 6
**Confidence:** 3

**Summary:**

This paper, "HWC-LOCO," proposes a novel hierarchical whole-body control (HWC) approach for robust humanoid locomotion. The core contribution is a hierarchical policy designed to dynamically resolve the trade-off between aggressive goal-tracking and conservative safety-recovery, particularly under environmental disturbances and Sim2Real mismatch. The authors frame the policy learning as a robust constrained reinforcement learning (CRL) problem, maximizing task rewards while ensuring worst-case feasibility constraints across an uncertainty set of transition dynamics. The hierarchical structure consists of a high-level planner that switches between a task-oriented goal-tracking policy (trained with a mimic learning objective for natural, human-like motion) and a stability-focused safety-recovery policy (enforcing ZMP-based constraints). The method is evaluated extensively in simulation and on a real-world humanoid platform, demonstrating superior robustness and performance compared to state-of-the-art baselines across various terrains and

**Strengths:**

The paper presents a novel and well-motivated combination of existing concepts. The formulation of humanoid locomotion as a robust constrained RL problem (Equation 4) is a significant and original contribution, moving beyond simple reward-shaping for safety. The hierarchical architecture, which explicitly and dynamically switches between a task-maximization mode and a safety-guarantee mode, is a practical and elegant solution to the classic robustness-vs-performance trade-off in safety-critical systems.

This work is highly significant for the field of humanoid robotics. Robust and reliable locomotion is a critical bottleneck for real-world deployment. By providing a principled way to integrate safety guarantees (via ZMP-based constraints in the recovery policy) with high-performance task execution (via the goal-tracking policy), HWC-Loco offers a foundational advancement. The demonstrated Sim2Real success and superior robustness under external disturbances suggest a practical and deployable control framework.

**Weaknesses:**

1. While the overall formulation is novel, the concept of a hierarchical controller switching between a task-policy and a recovery-policy is not entirely new in robotics (e.g., in model-based control or even some prior RL works). The paper's novelty rests heavily on the robust constrained RL formulation that trains this hierarchy. The authors should more explicitly discuss and contrast their hierarchical training approach with prior hierarchical execution methods to better highlight the distinction.

2. The robust optimization objective (Eq. 4) depends on the uncertainty set parameter $\alpha$ and the feasibility constraint $\epsilon$. The paper mentions that $\alpha$ specifies the scale of mismatch, but the sensitivity of the final policy's performance and robustness to the choice of $\alpha$ is not thoroughly explored. A poor choice of $\alpha$ could lead to either an overly conservative or insufficiently robust policy. More detailed analysis or guidance on selecting these critical parameters would strengthen the work.

3. Robust RL methods, especially those involving a max-min objective or sampling from an uncertainty set of dynamics, are typically computationally expensive. The paper does not provide sufficient detail on the training time or computational resources required for HWC-Loco compared to the baseline methods. Given the complexity of whole-body humanoid control, this is a crucial practical consideration that should be addressed.

**Questions:**

1. the paper compares HWC-Loco against standard baselines. Could the authors provide an ablation study comparing the full HWC-Loco (trained with Robust CRL, Eq. 4) against a version trained with the simpler Constrained RL (Eq. 2) or even a standard RL with a large penalty for constraint violation? This would isolate the benefit of the max-min robust optimization component specifically.

2. The high-level planner is key to the dynamic trade-off. What is the exact trigger mechanism for switching from the goal-tracking policy to the safety-recovery policy? Is it a simple threshold on a stability metric (e.g., ZMP distance from the support polygon), or is it a learned policy itself? Please elaborate on the input features and the decision logic of the planner.

3. The safety-recovery policy enforces ZMP-based constraints. Is this policy trained to be general across all tasks and terrains, or is it specialized? If it is a fixed, model-based controller, please state this clearly. If it is a learned policy, how is its robustness guaranteed, and how does it interact with the goal-tracking policy's learned dynamics?

4. The Sim2Real success is impressive. Were any specific system identification or domain randomization techniques used to tune the simulation parameters to match the real robot before training? If so, please detail these steps, as they are often critical for successful Sim2Real transfer in complex systems like humanoids.

---

> ### Author Response · Authors · 2025-11-22
>
> Dear Reviewer L28V, we truly value the time and effort you've dedicated to reviewing our work. In response, we have provided detailed clarifications with the aim of addressing your concerns.
>
> ---
> >1. W1: Novelty of the hierarchical structure.
> ---
> Thanks for raising this point. Our contribution lies in **how the hierarchy is trained and coordinated** under a robust constrained RL framework. π₁ is trained with constrained RL + Wasserstein imitation for human-like tracking, π₂ with a worst-case robust objective over an uncertainty set while enforcing ZMP-based feasibility, and π₀ as a Double-DQN that decides when to invoke π₂ from stability-related observations instead of hand-tuned thresholds. A heuristic ZMP-threshold baseline (Appendix B.2, Tables 17–20) shows that the learned planner π₀ achieves a better robustness–tracking–intervention trade-off than rule-based switching.
>
> ---
> >2. W2: Sensitivity to the uncertainty set parameter.
> ---
> Thanks for raising this point. In our implementation, α is not an abstract tuning knob: it directly parameterizes ranges of physically meaningful quantities (e.g., mass and payload, friction coefficients, external forces/torques, action delay, and sensor noise). These ranges are derived from measured hardware properties and conservative engineering margins, so that the uncertainty set captures realistic Sim2Real mismatch without introducing unphysical dynamics.
>
> We scale the disturbance and parameter ranges by {0.5, 1.0, 2.0}, with the original setting at scale = 1.0, and observe the following effects: smaller α improves nominal tracking but reduces robustness, while larger α leads to more conservative but robust behavior. The α value used in the main experiments represents a balance, chosen based on a held-out disturbance validation set. The quantitative effects are summarized below:
>
> | **Policy**| **Success Rate** ↑ | **Goal-tracking** ↑ | **Human-like** ↓ |
> |-|-|-|-|
> | HWC-Loco (0.5)| 97.88 ± 0.26| **1.14 ± 0.24**| **3.10 ± 0.11**|
> | HWC-Loco (1.0)| 98.14 ± 0.35| 1.12 ± 0.01| 3.11 ± 0.03|
> | HWC-Loco (2.0)| **98.20 ± 0.33**| 1.10 ± 0.05| 3.12 ± 0.14|
>
>
> ---
> >3. W3: Computational cost and scalability.
> ---
> Thanks for raising this point. We agree that training cost is practically important. In the revised manuscript, we report the training time for HWC-Loco and other baselines, so that the computational overhead of our robust formulation can be assessed more transparently.
>
> ---
> >4. Q1: Ablation for non-robust constrained RL and penalty-based baselines.
> ---
>
> Thanks for this insightful comment. To isolate the benefit of the max–min robust optimization component in Eq. (4), we compare HWC-Loco against:
>
> - **CRL**: a constrained RL variant that optimizes Eq. (2) using the same safety constraint but *without* the adversarial extreme set.
> - **RL-Penalty**: a standard RL baseline that uses a large penalty reward, combined with the same domain randomization as CRL.
>
> The results are:
>
> | **Policy**   | **Success Rate** ↑ | **Goal-tracking** ↑ | **Human-like** ↓ |
> |-|-|-|-|
> | CRL          | 98.12 ± 0.32      | 1.06 ± 0.08         | 3.56 ± 0.14      |
> | RL-Penalty   | 96.24 ± 0.13      | 1.11 ± 0.18         | 3.45 ± 0.23      |
> | **HWC-Loco** | **98.14 ± 0.35**  | **1.12 ± 0.01**     | **3.11 ± 0.03**  |
>
> As shown in the table, both CRL and RL-Penalty either sacrifice tracking performance or produce less natural, over-regularized gaits. In contrast, the full robust CRL formulation used by HWC-Loco achieves comparable success rates while delivering better command tracking and more human-like motion. This ablation confirms that the max–min robust optimization in Eq. (4) offers a clear advantage over standard constrained RL and penalty-based reward shaping.
>
> ---
> >5. Q2: High-level planner and switching mechanism.
> ---
> Thanks for this comment. The high-level planner $ \pi_0 $ is a **learned Double-DQN** policy that takes proprioceptive observations, VAE-estimated stability encodings (including ZMP features), and the current velocity command, outputting a discrete choice between $ \pi_1 $ and $ \pi_2 $. It maximizes task reward with penalties on switching and episode termination. A switching-strategy ablation (Appendix B.2) shows that the learned planner achieves a better robustness–tracking trade-off than a simple ZMP-distance threshold.

---

> > ### Author Response · Authors · 2025-11-22
> >
> > ---
> > >6. Q3: Nature and generality of the safety-recovery policy.
> > ---
> > Thanks for raising this point. The safety-recovery policy π₂ is a **learned RL policy** trained under the robust objective (Eq. 4) across all terrains in our experiments, with randomized commands and extensive dynamics/disturbance randomization.
> >
> > The ZMP-based feasibility term φ(τ) depends only on contact geometry and whole-body dynamics, making π₂ a **task-agnostic** recovery policy reused for all tasks and terrains in simulation and on hardware. π₁ and π₂ share observation and action spaces for stable switching, and a low-pass filter on joint targets further smooths transitions on the real robot (Appendix A.9).
> >
> > ---
> > >7. Q4: System identification and domain randomization for Sim2Real.
> > ---
> > Thanks for raising this point. Before training, we construct a baseline physics model for each humanoid from the manufacturer URDF (collision geometry, link masses, inertias, joint limits) and choose initial PD gains, control frequencies, and parameter ranges using existing humanoid RL frameworks (e.g., Unitree RL Gym) as guidance. We then run sim-to-sim validation in MuJoCo to ensure stable gaits before large-scale training in IsaacGym; domain-randomization ranges (Appendix A.6, Table 11) are chosen based on these libraries and a small set of calibration experiments on the real robots.

---

### Author Response · Authors · 2025-12-03
**Summary of updates - Part 1**

We sincerely thank the AC and the reviewers for their time and thoughtful feedback. This paper studies robust whole-body humanoid locomotion over diverse terrains and under strong disturbances, and proposes **HWC-Loco**, a hierarchical framework in which a goal-tracking policy, a task-agnostic safety-recovery policy, and a learned high-level switching policy are trained under a robust constrained RL formulation to jointly achieve efficiency, robustness, and natural motion.

We summarize the four reviews (L28V, qydQ, qF8c, tg3Q); the key issues can be grouped as follows:

1. **Novelty and concrete role of the robust constrained RL formulation and switching mechanism.** *(Reviewer L28V, qydQ)*

   We provided detailed clarification on our robust constrained formulation. The contribution lies not only in using a hierarchical architecture, but in **how the components are trained and coordinated**. The goal-tracking policy π₁ is optimized with constrained RL and Wasserstein imitation, balancing task performance and human-like motion. The safety-recovery policy π₂ is trained over a physically grounded uncertainty set together with a ZMP-based feasibility criterion. The high-level policy π₀ is a Double-DQN agent that operates on stability-related observations and learns when to invoke recovery, rather than relying on heuristic switching rules.

   To make this more concrete, we conducted ablations against both a non-robust constrained RL variant and a penalty-based RL baseline. These experiments show that **only the full robust formulation** consistently maintains high overall performance (success rate, tracking quality, and motion naturalness), thereby clarifying the specific contribution of each component. The additional results are reported in Appendix B.5 in the revised version (Non-robust constrained ablation).

2. **Sensitivity to design choices and robustness to estimation noise.** *(Reviewer L28V, qF8c, qydQ)*

   We conducted sensitivity analyses over both the uncertainty-set scale α and the imitation weight λ. Varying α produces a controlled trade-off between nominal performance and robustness, while variations in λ mainly affect gait naturalness and have limited impact on overall robustness, indicating that the method is not overly sensitive to these hyperparameters. The sensitivity analyses are reported in Appendix B.2 in the revised version.

   To address concerns about noise in VAE-based privileged features used by π₀, we clarified that the way we train π₀ is on noisy estimates and include an explicit penalty on excessive switching. We also added additional quantitative analyses that show low estimation errors and high success rates even under realistic injected noise levels, with only moderate increases in the number of switches, suggesting that the switching policy is robust to estimation noise. The estimation robustness analyses are reported in Appendix B.4 in the revised version.

3. **Necessity and practical impact of the hierarchical design vs. strong domain-randomized policies.** *(Reviewer L28V, qF8c, tg3Q)*

    To address those concerns, we considered both **strong DR baselines** and **recent state-of-the-art locomotion policies**:

    - For **Standard DR policies**, we introduce a Large-DR history-aware baseline that is trained with extensive domain randomization comparable to the safety-recovery policy. As randomization strength increases, this baseline tends to become overly conservative and less human-like, while HWC-Loco maintains comparable nominal performance but exhibits **substantially higher success rates and fewer catastrophic failures** under extreme perturbations.

    - For **recent SOTA methods**, we adapt Narrow[1] and Perceptive[2] in simulation and observe that HWC-Loco attains competitive or better success rates across diverse terrains and disturbance types, especially in the more challenging settings highlighted in the rebuttal. Across these tests, HWC-Loco more frequently recovers from disturbances and avoids falls, while the SOTA baselines are more prone to entering irrecoverable states.

    These results suggest that the hierarchical design is not merely comparable to strong DR and recent SOTA policies in nominal conditions, but provides **practically meaningful robustness gains** by reducing catastrophic failures and facilitating more reliable recovery—precisely in the **safety-critical regimes**. Appendix B.3 reports the results. Besides, we deployed HWC-Loco on **Unitree G1 humanoid** and provided additional demos (e.g., Single Leg Balance) on our anonymous project website ([HWC-Loco Demo](https://anonymous.4open.science/w/HWC_Loco_demo/)), with corresponding details in Appendix B.7 (Robust Balance under Disturbances).

---

> ### Author Response · Authors · 2025-12-03
> **Summary of updates - Part 2**
>
> **Recognized strengths of the work (as noted by the reviewers).**
>
> 1. **Well-motivated and principled formulation.** *(L28V, qydQ, qF8c)*
>    Reviewers note that the paper tackles an important and challenging problem—robust real-world humanoid locomotion—and that casting it as a *robust constrained RL* problem with an explicit safety/performance hierarchy is natural and conceptually well grounded.
>
> 2. **Significance for humanoid robotics and safety.** *(Reviewer L28V, qydQ)*
>    The work is considered highly relevant for real-world humanoid deployment, providing a principled way to combine ZMP-based safety constraints and a recovery policy with high-performance goal tracking in a unified framework.
>
> 3. **Effective hierarchical design and switching behavior.** *(Reviewer L28V, qF8c, tg3Q)*
>    The high-level policy and hierarchical architecture are regarded as well designed, with learned switching that empirically balances goal tracking and safety recovery and maintains safety without overly sacrificing performance.
>
> 4. **Extensive and strong experimental validation.** *(Reviewer qF8c, qydQ, tg3Q)*
>    Reviewers appreciate the extensive experiments across diverse terrains, commands, and disturbances, with detailed baselines, ablations, and metrics for both robustness and motion quality.
>
> 5. **Sim2Real performance and practical deployability.** *(Reviewer L28V, qydQ, tg3Q)*
>    The demonstrated Sim2Real results and improved robustness under external disturbances are seen as strong indications that HWC-Loco is practically applicable and a promising framework for robust humanoid locomotion.
>
> **Post-rebuttal reviewer perspectives.** According to the reviewers’ feedback, Reviewer **qF8c** confirmed that our response successfully addressed the previous concerns and **raised the score**. Reviewer **qydQ** (initial score **8**) **maintained a high evaluation** and noted that the rebuttal “brings further clarity” to the paper. Reviewer **L28V** (initial score **6**) also **retained a positive assessment** without introducing any new critical issues. Reviewer **tg3Q** raised **additional questions** regarding whether the improvements over strong DR baselines are substantial and how well the approach generalizes to real-world deployment. In response, we provided more **detailed clarifications of the experimental settings and introduced additional experiments in both simulation and real-world environments**. Although the issue with the OpenReview system affected the discussion, we believe our response effectively resolves the reviewers’ concerns.
>
> Overall, the additional clarifications, experiments, and analyses further clarify the problem setting, highlight the novelty of our method, and strengthen the empirical support for HWC-Loco. We sincerely hope this summary helps the AC in evaluating our work.
>
> **References:**
> [1] Xie W, Bai C, Shi J, et al. *“Humanoid Whole-Body Locomotion on Narrow Terrain via Dynamic Balance and Reinforcement Learning.”* arXiv, 2025.
> [2] Sun W, Cao B, Chen L, et al. *“Learning Perceptive Humanoid Locomotion over Challenging Terrain.”* arXiv, 2025.

---

### Meta-Review · Area_Chair_98fr · 2026-01-05

**Summary:**

This paper proposes HWC-Loco, a hierarchical whole-body control framework for robust humanoid locomotion. The method uses a high-level planner to dynamically switch between a goal-tracking policy (optimized for performance and human-like motion) and a safety-recovery policy (trained under a robust constrained RL formulation to handle disturbances and model mismatch). However, the novelty of the hierarchical structure was challenged; a clearer visualization of switching behavior and more comprehensive comparisons with strong recent baselines can further strengthen this work.

**Reviewer Concerns:**

For reviewers L28V and qydQ, the concerns are such as framework complexity/reproducibility, potential chattering from noisy VAE estimates, sensitivity to the uncertainty set parameter (α), computational cost, and requested ablations vs. non-robust constrained RL. For reviewer qF8c, the paper is missing comparisons to strong recent baselines. Those concerns were partially addressed by additional experiments. For reviewer tg3Q, the paper needs a clearer visualization of switching behavior and limitations of ZMP-based constraints for highly dynamic motions. These concerns were partially addressed.

**Reviewer Scores:**

Three reviewers initially at or below the acceptance threshold (4, 4, 6) were positively influenced by the rebuttal, with two explicitly or likely raising their scores. The high-scoring reviewer (8) remained strongly positive. Overall, the strong rebuttal gets the paper over the bar for acceptance.

---

### Decision · Program_Chairs · 2026-01-26

Accept (Poster)